# Pretraining Decision Transformers with Reward Prediction for In-Context Multi-task Structured Bandit Learning

## Abstract

In this paper, we study the multi-task structured bandit problem where the goal is to learn a near-optimal algorithm that minimizes cumulative regret. The tasks share a common structure and the algorithm exploits the shared structure to minimize the cumulative regret for an unseen but related test task. We use a transformer as a decision-making algorithm to learn this shared structure so as to generalize to the test task. The prior work of pretrained decision transformers like DPT requires access to the optimal action during training which may be hard in several scenarios. Diverging from these works, our learning algorithm does not need the knowledge of optimal action per task during training but predicts a reward vector for each of the actions using only the observed offline data from the diverse training tasks. Finally, during inference time, it selects action using the reward predictions employing various exploration strategies in-context for an unseen test task. We show that our model outperforms other SOTA methods like DPT, and Algorithmic Distillation (AD) over a series of experiments on several structured bandit problems (linear, bilinear, latent, non-linear). Interestingly, we show that our algorithm, without the knowledge of the underlying problem structure, can learn a near-optimal policy in-context by leveraging the shared structure across diverse tasks. We further extend the field of pre-trained decision transformers by showing that they can leverage unseen tasks with new actions and still learn the underlying latent structure to derive a near-optimal policy. We validate this over several experiments to show that our proposed solution is very general and has wide applications to potentially emergent online and offline strategies at test time. Finally, we theoretically analyze the performance of our algorithm and obtain generalization bounds in the in-context multi-task learning setting.

## 1 Introduction

In this paper, we study multi-task bandit learning with the goal of learning an algorithm that discovers and exploits structure in a family of related tasks. In multi-task bandit learning, we have multiple distinct bandit tasks for which we want to learn a policy. Though distinct, the tasks share some structure, which we hope to leverage to speed up learning on new instances in this task family. Traditionally, the study of such structured bandit problems has relied on knowledge of the problem structure like linear bandits (Li et al., 2010; Abbasi-Yadkori et al., 2011; Degenne et al., 2020), bilinear bandits (Jun et al., 2019), hierarchical bandits (Hong et al., 2022a;b), Lipschitz bandits (Bubeck et al., 2008; 2011; Magureanu et al., 2014), other structured bandits settings (Riquelme et al., 2018; Lattimore & Szepesvári, 2019; Dong et al., 2021) and even linear and bilinear multi-task bandit settings (Yang et al., 2022a; Du et al., 2023; Mukherjee et al., 2023). When structure is unknown an alternative is to adopt sophisticated model classes, such as kernel machines or neural networks, exemplified by kernel or neural bandits (Valko et al., 2013; Chowdhury & Gopalan, 2017; Zhou et al., 2020; Dai et al., 2022). However, these approaches are also costly as they learn complex, nonlinear models from the ground up without any prior data (Justus et al., 2018; Zambaldi et al., 2018).

In this paper, we consider an alternative approach of synthesizing a bandit algorithm from historical data where the data comes from recorded bandit interactions with past instances of our target task family. Concretely, we are given a set of state-action-reward tuples obtained by running some bandit

algorithm in various instances from the task family. We then aim to train a transformer (Vaswani et al., 2017) from this data such that it can learn in-context to solve new task instances. Laskin et al. (2022) consider a similar goal and introduce the Algorithm Distillation (AD) method, however, AD aims to copy the algorithm used in the historical data and thus is limited by the ability of the data collection algorithm. Lee et al. (2023) develop an approach, DPT, that enables learning a transformer that obtains lower regret in-context bandit learning compared to the algorithm used to produce the historical data. However, this approach requires knowledge of the optimal action at each stage of the decision process. In real problems, this assumption is hard to satisfy and we will show that DPT performs poorly when the optimal action is only approximately known. With this past work in mind, the goal of this paper is to answer the question:

*Can we learn an in-context bandit learning algorithm that obtains lower regret than the algorithm used to produce the training data without knowledge of the optimal action in each training task?*

To answer this question, we introduce a new pre-training methodology, called **Pre**-trained **De**cision **T**ransf**o**rmer with **R**eward Estimation (PreDeToR) that obviates the need for knowledge of the optimal action in the in-context data — a piece of information that is often inaccessible. Our key observation is that while the mean rewards of each action change from task to task, certain probabilistic dependencies are persistent across all tasks with a given structure (Yang et al., 2020; 2022a; Mukherjee et al., 2023). These probabilistic dependencies can be learned from the pretraining data and exploited to better estimate mean rewards and improve performance in a new unknown test task. The nature of the probabilistic dependencies depends on the specific structure of the bandit and can be complex (i.e., higher-order dependencies beyond simple correlations). We propose to use transformer models as a general-purpose architecture to capture the unknown dependencies by training transformers to predict the mean rewards in each of the given trajectories (Mirchandani et al., 2023; Zhao et al., 2023). The key idea is that transformers have the capacity to discover and exploit complex dependencies in order to predict the rewards of all possible actions in each task from a *small* history of action-reward pairs in a new task. This paper demonstrates how such an approach can achieve lower regret by outperforming state-of-the-art baselines, relying solely on historical data, without the need for any supplementary information like the action features or knowledge of the complex reward models. We also show that the shared actions across the tasks are vital for PreDeToR to exploit the latent structure. We show that PreDeToR learns to adapt, in-context, to novel actions and new tasks as long as the number of new actions is small compared to shared actions across the tasks.

**Contributions**

1. We introduce a new pre-training procedure of learning the underlying reward structure and a decision algorithm. Moreover, PreDeToR by predicting the next reward for all arms circumvents the issue of requiring access to the optimal (or approximately optimal) action during training time.

2. We demonstrate empirically that this training procedure results in lower regret in a wide series of tasks (such as linear, nonlinear, bilinear, and latent bandits) compared to prior in-context learning algorithms and bandit algorithms with privileged knowledge of the common structure.

3. We also show that our training procedure leverages the shared latent structure and is robust to a small number of new actions introduced both during training and testing time.

4. Finally, we theoretically analyze the generalization ability of PreDeToR through the lens of algorithmic stability and new results for the transformer setting.

## 2 BACKGROUND

In this section, we first introduce our notation and the multi-task, structured bandit setting. We then formalize the in-context bandit learning model studied in Laskin et al. (2022); Lee et al. (2023); Sinii et al. (2023); Lin et al. (2023); Ma et al. (2023); Liu et al. (2023c;a).

## 2.1 PRELIMINARIES

In this paper, we consider the multi-task linear bandit setting (Du et al., 2023; Yang et al., 2020; 2022a). In the multi-task setting, we have a family of related bandit problems that share an action set $\mathcal{A}$ and also a common action feature space $\mathcal{X}$. The actions in $\mathcal{A}$ are indexed by $a = 1, 2, \ldots, A$. The feature of each action is denoted by $\mathbf{x}(a) \in \mathbb{R}^d$ and $d \ll A$. A policy, $\pi$, is a probability distribution over the actions.

Define $[n] = \{1, 2, \ldots, n\}$. In a multi-task structured bandit setting the expected reward for each action in each task is assumed to be an unknown function of the hidden parameter and action features (Lattimore & Szepesvári, 2020; Gupta et al., 2020). The interaction proceeds iteratively over $n$ rounds for each task $m \in [M]$. At each round $t \in [n]$ for each task $m \in [M]$, the learner selects an action $I_{m,t} \in \mathcal{A}$ and observes the reward $r_{m,t} = f(\mathbf{x}(I_{m,t}), \boldsymbol{\theta}_{m,*}) + \eta_{m,t}$, where $\boldsymbol{\theta}_{m,*} \in \mathbb{R}^d$ is the hidden parameter specific to the task $m$ to be learned by the learner. The function $f(\cdot, \cdot)$ is the unknown reward structure. This can be $f(\mathbf{x}(I_{m,t}), \boldsymbol{\theta}_{m,*}) = \mathbf{x}(I_{m,t})^\top \boldsymbol{\theta}_{m,*}$ for the linear setting or even more complex correlation between features and $\boldsymbol{\theta}_{m,*}$ (Filippi et al., 2010; Abbasi-Yadkori et al., 2011; Riquelme et al., 2018; Lattimore & Szepesvári, 2019; Dong et al., 2021).

In our paper, we assume that there exist weak demonstrators denoted by $\pi^w$. These weak demonstrators are stochastic $A$-armed bandit algorithms like Upper Confidence Bound (UCB) (Auer et al., 2002; Auer & Ortner, 2010) or Thompson Sampling (Thompson, 1933; Agrawal & Goyal, 2012; Russo et al., 2018; Zhu & Tan, 2020). We refer to these algorithms as weak demonstrators because they do not use knowledge of task structure or arm feature vectors to plan their sampling policy. In contrast to a weak demonstrator, a strong demonstrator, like LinUCB, uses feature vectors and knowledge of task structure to conduct informative exploration. Whereas weak demonstrators always exist, there are many real-world settings with no known strong demonstrator algorithm or where the feature vectors are unobserved and the learner can only use the history of rewards and actions.

## 2.2 IN-CONTEXT LEARNING MODEL

Similar to Lee et al. (2023); Sinii et al. (2023); Lin et al. (2023); Ma et al. (2023); Liu et al. (2023c;a) we assume the in-context learning model. We first discuss the pretraining procedure.

**Pretraining:** Let $\mathcal{T}_{\text{pre}}$ denote the distribution over tasks $m$ at the time of pretraining. Let $\mathcal{D}_{\text{pre}}$ be the distribution over all possible interactions that the $\pi^w$ can generate. We first sample a task $m \sim \mathcal{T}_{\text{pre}}$ and then a context $\mathcal{H}_m$ which is a sequence of interactions for $n$ rounds conditioned on the task $m$ such that $\mathcal{H}_m \sim \mathcal{D}_{\text{pre}}(\cdot|m)$. So $\mathcal{H}_m = \{I_{m,t}, r_{m,t}\}_{t=1}^n$. We call this dataset $\mathcal{H}_m$ an in-context dataset as it contains the contextual information about the task $m$. We denote the samples in $\mathcal{H}_m$ till round $t$ as $\mathcal{H}_m^t = \{I_{m,s}, r_{m,s}\}_{s=1}^{t-1}$. This dataset $\mathcal{H}_m$ can be collected in several ways: (1) random interactions within $m$, (2) demonstrations from an expert, and (3) rollouts of an algorithm. Finally, we train a causal GPT-2 transformer model TF parameterized by $\boldsymbol{\Theta}$ on this dataset $\mathcal{D}_{\text{pre}}$. Specifically, we define $\text{TF}_{\boldsymbol{\Theta}}(\cdot \mid \mathcal{H}_m^t)$ as the transformer model that observes the dataset $\mathcal{H}_m^t$ till round $t$ and then produces a distribution over the actions. Our primary novelty lies in our training procedure which we explain in detail in Section 3.1.

**Testing:** We now discuss the testing procedure for our setting. Let $\mathcal{T}_{\text{test}}$ denote the distribution over test tasks $m \in [M_{\text{test}}]$ at the time of testing. Let $\mathcal{D}_{\text{test}}$ denote a distribution over all possible interactions that can be generated by $\pi^w$ during test time. At deployment time, the dataset $\mathcal{H}_m^0 \leftarrow \{\emptyset\}$ is initialized empty. At each round $t$, an action is sampled from the trained transformer model $I_t \sim \text{TF}_{\boldsymbol{\Theta}}(\cdot \mid \mathcal{H}_m^t)$. The sampled action and resulting reward, $r_t$, are then added to $\mathcal{H}_m^t$ to form $\mathcal{H}_m^{t+1}$ and the process repeats for $n$ total rounds. Finally, note that in this testing phase, the model parameter $\boldsymbol{\Theta}$ is not updated. Finally, the goal of the learner is to minimize cumulative regret for all task $m \in [M_{\text{test}}]$ defined as follows: $\mathbb{E}[R_n] = \frac{1}{M_{\text{test}}} \sum_{m=1}^{M_{\text{test}}} \sum_{t=1}^n \max_{a \in \mathcal{A}} f(\mathbf{x}(a), \boldsymbol{\theta}_{m,*}) - f(\mathbf{x}(I_t), \boldsymbol{\theta}_{m,*})$.

## 2.3 RELATED IN-CONTEXT LEARNING ALGORITHMS

In this section, we discuss related algorithms for in-context decision-making. For completeness, we describe the DPT and AD training procedure and algorithm now. During training, DPT first samples $m \sim \mathcal{T}_{\text{pre}}$ and then an in-context dataset $\mathcal{H}_m \sim \mathcal{D}_{\text{pre}}(\cdot|, m)$. It adds this $\mathcal{H}_m$ to the training dataset $\mathcal{H}_{\text{train}}$, and repeats to collect $M_{\text{pre}}$ such training tasks. For each task $m$, DPT requires the optimal action $a_{m,*} = \arg\max_a f(\mathbf{x}(m, a), \boldsymbol{\theta}_{m,*})$ where $f(\mathbf{x}(m, a), \boldsymbol{\theta}_{m,*})$ is the expected reward for the action $a$ in task $m$. Since the optimal action is usually not known in advance, in Section 4 we introduce a practical variant of DPT that approximates the optimal action with the best action

identified during task interaction. During training DPT minimizes the cross-entropy loss:

$$\mathcal{L}_t^{\text{DPT}} = \text{cross-entropy}(\text{TF}_{\boldsymbol{\Theta}}(\cdot|\mathcal{H}_m^t), p(a_{m,*})) \tag{1}$$

where $p(a_{m,*}) \in \triangle^{\mathcal{A}}$ is a one-hot vector such that $p(j) = 1$ when $j = a_{m,*}$ and 0 otherwise. This loss is then back-propagated and used to update the model parameter $\boldsymbol{\Theta}$.

During test time evaluation for online setting the DPT selects $I_t \sim \text{softmax}_a^\tau(\text{TF}_{\boldsymbol{\Theta}}(\cdot|\mathcal{H}_m^t))$ where we define the $\text{softmax}_a^\tau(\mathbf{v})$ over a $A$ dimensional vector $\mathbf{v} \in \mathbb{R}^A$ as $\text{softmax}_a^\tau(\mathbf{v}(a)) = \exp(\mathbf{v}(a)/\tau)/\sum_{a'=1}^A \exp(\mathbf{v}(a')/\tau)$ which produces a distribution over actions weighted by the temperature parameter $\tau > 0$. Therefore this sampling procedure has a high probability of choosing the predicted optimal action as well as induce sufficient exploration. In the online setting, the DPT observes the reward $r_t(I_t)$ which is added to $\mathcal{H}_m^t$. So the $\mathcal{H}_m$ during online testing consists of $\{I_t, r_t\}_{t=1}^n$ collected during testing. This interaction procedure is conducted for each test task $m \in [M_{\text{test}}]$. In the testing phase, the model parameter $\boldsymbol{\Theta}$ is not updated.

An alternative to DPT that does *not* require knowledge of the optimal action is the AD approach (Laskin et al., 2022; Lu et al., 2023). In AD, the learner aims to predict the next action of the demonstrator. So it minimizes the cross-entropy loss as follows:

$$\mathcal{L}_t^{\text{AD}} = \text{cross-entropy}(\text{TF}_{\boldsymbol{\Theta}}(\cdot|\mathcal{H}_m^t), p(I_{m,t})) \tag{2}$$

where $p(I_{m,t})$ is a one-hot vector such that $p(j) = 1$ when $j = I_{m,t}$ (the true action taken by the demonstrator) and 0 otherwise. At deployment time, AD selects $I_t \sim \text{softmax}_a^\tau(\text{TF}_{\boldsymbol{\Theta}}(\cdot|\mathcal{H}_m^t))$. Note that the objective of AD is to match the performance of the demonstrator. In the next section, we introduce a new method that can improve upon the demonstrator without knowledge of the optimal action.

## 3 PROPOSED ALGORITHM PREDETOR

We now introduce our main algorithmic contribution, PreDeToR (which stands for **Pre**-trained **De**cision **T**ransf**o**rmer with **R**eward Estimation).

### 3.1 PRE-TRAINING NEXT REWARD PREDICTION

The key idea behind PreDeToR is to leverage the in-context learning ability of transformers to infer the reward of each arm in a given test task. By training this in-context ability on a set of training tasks, the transformer can implicitly learn structure in the task family and exploit this structure to infer rewards without trying every single arm. Thus, in contrast to DPT and AD that output actions directly, PreDeToR outputs a scalar value reward prediction for each arm. To this effect, we append a linear layer of dimension $A$ on top of a causal GPT2 model, denoted by $\text{TF}^{\mathbf{r}}_{\boldsymbol{\Theta}}(\cdot|\mathcal{H}_m)$, and use a least-squares loss to train the transformer to predict the reward for each action with these outputs. Note that we use $\text{TF}^{\mathbf{r}}_{\boldsymbol{\Theta}}(\cdot|\mathcal{H}_m)$ to denote a reward prediction transformer and $\text{TF}_{\boldsymbol{\Theta}}(\cdot|\mathcal{H}_m)$ as the transformer that predicts a distribution over actions (as in DPT and AD ). At every round $t$ the transformer predicts the *next reward* for each of the actions $a \in \mathcal{A}$ for the task $m$ based on $\mathcal{H}_m^t = \{I_{m,s}, r_{m,s}\}_{s=1}^{t-1}$. This predicted reward is denoted by $\widehat{r}_{m,t+1}(a)$ for each $a \in \mathcal{A}$.

**Loss calculation:** For each training task, $m$, we calculate the loss at each round, $t$, using the transformer's prediction $\hat{r}_{m,t}(I_{m,t})$ and the actual observed reward $r_{m,t}$ that followed action $I_{m,t}$. We use a least-squares loss function:

$$\mathcal{L}_t = (\widehat{r}_{m,t}(I_{m,t}) - r_{m,t})^2 \tag{3}$$

and hence minimizing this loss will minimize the mean squared-error of the transformer's predictions. The loss is calculated using equation 3 and is backpropagated to update the model parameter $\boldsymbol{\Theta}$.

**Exploratory Demonstrator:** Observe from the loss definition in equation 3 that it is calculated from the observed true reward and action from the dataset $\mathcal{H}_m$. In order for the transformer to learn accurate reward predictions during training, we require that the weak demonstrator is sufficiently exploratory such that it collects $\mathcal{H}_m$ such that $\mathcal{H}_m$ contains some reward $r_{m,t}$ for each action $a$. We discuss in detail the impact of the demonstrator on PreDeToR (-$\tau$) training in Appendix A.13.

## 3.2 Deploying PreDeToR

At deployment time, PreDeToR learns in-context to predict the mean reward of each arm on an unseen task and acts greedily with respect to this prediction. That is, at deployment time, a new task is sampled, $m \sim \mathcal{T}_{\text{test}}$, and the dataset $\mathcal{H}_m^0$ is initialized empty. Then at every round $t$, PreDeToR chooses $I_t = \arg\max_{a \in \mathcal{A}} \text{TF}^{\mathbf{r}}_{\Theta} (\widehat{r}_{m,t}(a) \mid \mathcal{H}_m^t)$ which is the action with the highest predicted reward and $\widehat{r}_{m,t}(a)$ is the predicted reward of action $a$. Note that PreDeToR is a greedy policy and thus may fail to conduct sufficient exploration. To remedy this potential limitation, we also introduce a soft variant, PreDeToR-$\tau$ that chooses $I_t \sim \text{softmax}_a^\tau (\text{TF}^{\mathbf{r}}_{\Theta} (\widehat{r}_{m,t}(a) \mid \mathcal{H}_m^t))$. For both PreDeToR and PreDeToR-$\tau$, the observed reward $r_t(I_t)$ is added to the dataset $\mathcal{H}_m$ and then used to predict the reward at the next round $t + 1$. The full pseudocode of using PreDeToR for online interaction is shown in Algorithm 1. In Appendix A.15, we discuss how PreDeToR (-$\tau$) can be deployed for offline learning.

---

**Algorithm 1 Pre-trained Decision Transformer with Reward Estimation (PreDeToR)**

---

1: **Collecting Pretraining Dataset**
2: Initialize empty pretraining dataset $\mathcal{H}_{\text{train}}$
3: **for** $i$ in $[M_{\text{pre}}]$ **do**
4:     Sample task $m \sim \mathcal{T}_{\text{pre}}$, in-context dataset $\mathcal{H}_m \sim \mathcal{D}_{\text{pre}}(\cdot|m)$ and add this to $\mathcal{H}_{\text{train}}$.
5: **end for**
6: **Pretraining model on dataset**
7: Initialize model $\text{TF}^{\mathbf{r}}_{\Theta}$ with parameters $\Theta$
8: **while** not converged **do**
9:     Sample $\mathcal{H}_m$ from $\mathcal{H}_{\text{train}}$ and predict $\widehat{r}_{m,t}$ for action $(I_{m,t})$ for all $t \in [n]$
10:     Compute loss in equation 3 with respect to $r_{m,t}$ and backpropagate to update model parameter $\Theta$.
11: **end while**
12: **Online test-time deployment**
13: Sample unknown task $m \sim \mathcal{T}_{\text{test}}$ and initialize empty $\mathcal{H}_m^0 = \{\emptyset\}$
14: **for** $t = 1, 2, \ldots, n$ **do**
15:     Use $\text{TF}^{\mathbf{r}}_{\Theta}$ on $m$ at round $t$ to choose

$$I_t \begin{cases} = \arg\max_{a \in \mathcal{A}} \text{TF}^{\mathbf{r}}_{\Theta} (\widehat{r}_{m,t}(a) \mid \mathcal{H}_m^t), & \textbf{PreDeToR} \\ \sim \text{softmax}_a^\tau \text{TF}^{\mathbf{r}}_{\Theta} (\widehat{r}_{m,t}(a) \mid \mathcal{H}_m^t), & \textbf{PreDeToR-}\tau \end{cases}$$

16:     Add $\{I_t, r_t\}$ to $\mathcal{H}_m^t$ to form $\mathcal{H}_m^{t+1}$.
17: **end for**

---

## 4 Empirical Study: Non-Linear Structure

Having introduced PreDeToR, we now investigate its performance in diverse bandit settings compared to other in-context learning algorithms. In our first set of experiments, we use a bandit setting with a common non-linear structure across tasks. Ideally, a good learner would leverage the structure, however, we choose the structure such that no existing algorithms are well-suited to the non-linear structure. This setting is thus a good testbed for establishing that in-context learning can discover and exploit common structure. Moreover, each task only consists of a few rounds of interactions. This setting is quite common in recommender settings where user interaction with the system lasts only for a few rounds and has an underlying non-linear structure (Kwon et al., 2022; Tomkins et al., 2020). We show that PreDeToR achieves lower regret than other in-context algorithms for the non-linear structured bandit setting. We study the performance of PreDeToR in the large horizon setting in Appendix A.7.

**Baselines:** We first discuss the baselines used in this setting.

**(1) PreDeToR:** This is our proposed method shown in Algorithm 1.

**(2) PreDeToR-$\tau$:** This is the proposed exploratory method shown in Algorithm 1 and we fix $\tau = 0.05$.

**(3) DPT-greedy:** This baseline is the greedy approximation of the DPT algorithm from Lee et al. (2023) which is discussed in Section 2.3. Note that we choose DPT-greedy as a representative

example of similar in-context decision-making algorithms studied in Lee et al. (2023); Sinii et al. (2023); Lin et al. (2023); Ma et al. (2023); Liu et al. (2023c;a) all of which require the optimal action (or its greedy approximation). DPT-greedy estimates the optimal arm using the reward estimates for each arm during each task.

**(4) AD:** This is the Algorithmic Distillation method (Laskin et al., 2022; Lu et al., 2021) discussed in Section 2.3.

**(5) Thomp:** This baseline is the celebrated stochastic $A$-action bandit Thompson Sampling algorithm from Thompson (1933); Agrawal & Goyal (2012); Russo et al. (2018); Zhu & Tan (2020). We choose Thomp as the weak demonstrator $\pi^w$ as it does not make use of arm features. Thomp is also a stochastic algorithm that induces more exploration in the demonstrations.

**(6) LinUCB:** (Linear Upper Confidence Bound): This baseline is the Upper Confidence Bound algorithm for the linear bandit setting that leverages the linear structure and feature of the arms to select the most promising action as well as conducting exploration. We choose LinUCB as a baseline for each test task to show the limitations of algorithms that use linear feedback structure as an underlying assumption to select actions. Note that LinUCB requires oracle access to features to select actions per task.

**(7) MLinGreedy:** This is the multi-task linear regression bandit algorithm proposed by Yang et al. (2021). This algorithm assumes that there is a common low-dimensional feature extractor shared between the tasks and the reward of each task linearly depends on this feature extractor. We choose MLinGreedy as a baseline to show the limitations of algorithms that use linear feedback structure *across tasks* as an underlying assumption to select actions. Note that MLinGreedy requires oracle access to the action features to select actions as opposed to DPT, AD, and PreDeToR.

We describe in detail the baselines Thomp, LinUCB, and MLinGreedy for interested readers in Appendix A.2.2.

**Outcomes:** Before presenting the result we discuss the main outcomes from our experimental results in this section:

> **Finding 1:** PreDeToR (-$\tau$) lowers regret compared to other baselines under unknown, non-linear structure. It learns to exploit the latent structure of the underlying tasks from in-context data even when it is trained without the optimal action $a_{m,*}$ (or its approximation) and without action features $\mathcal{X}$.

**Experimental Result:** These findings are reported in Figure 1. In Figure 1(a) we show the non-linear bandit setting for horizon $n = 50$, $M_{\text{pre}} = 100000$, $M_{\text{test}} = 200$, $A = 6$, and $d = 2$. The demonstrator $\pi^w$ is the Thomp algorithm. We observe that PreDeToR (-$\tau$) has lower cumulative regret than DPT-greedy. Note that for this low data regime (short horizon) the DPT-greedy does not have a good estimation of $\widehat{a}_{m,*}$ which results in a poor prediction of optimal action $\widehat{a}_{m,t,*}$. This results in higher regret. The PreDeToR (-$\tau$) has lower regret than LinUCB, and MLinGreedy, which fail to perform well in this non-linear setting due to their algorithmic design and linear feedback assumption. Finally, PreDeToR-$\tau$ performs slightly better than PreDeToR in both settings as it conducts more exploration.

In Figure 1(b) we show the non-linear bandit setting for horizon $n = 25$, $M_{\text{pre}} = 100000$, $M_{\text{test}} = 200$, $A = 6$, and $d = 2$ where the norm of the $\boldsymbol{\theta}_{m,*}$ determines the reward of the actions which also is a non-linear function $\boldsymbol{\theta}_{m,*}$ and action features. This setting is similar to the wheel bandit setting of Riquelme et al. (2018). Again, we observe that PreDeToR has lower cumulative regret than all the other baselines.

Finally in Figure 1(c) and Figure 1(d) we show the performance of PreDeToR against other baselines in real-world datasets Movielens and Yelp. The Movielens dataset consists of more than 32 million ratings of 200,000 users and 80,000 movies (Harper & Konstan, 2015) where each entry consists of user-id, movie-id, rating, and timestamp. The Yelp dataset (Asghar, 2016) consists of ratings of 1300 business categories by 150,000 users. Each entry is summarized as user-id, business-id, rating, and timestamp. Previously structured bandit works (Deshpande & Montanari, 2012; Hong et al., 2023) directly fit a linear structure or low-rank factorization to estimate the $\boldsymbol{\theta}_{m,*}$ and simulate the

ratings. However, we directly use the user-ids and movie-ids (or business-ids) to build a histogram of ratings per user and calculate the mean rating per movie (or business-id) per task. Define this as the $\{\mu_{m,a}\}_{a=1}^{A}$. This is then used to simulate the rating for $n$ horizon per movie per task where the data collection algorithm is uniform sampling. Note that this does not require estimation of user or movie features, and PreDeToR (-$\tau$) learns to exploit the latent structure of user-movie (or business) rating correlations directly from the data. From Figure 1(c) and Figure 1(d) we see that PreDeToR, and PreDeToR-$\tau$ outperform all the other baselines in these settings.

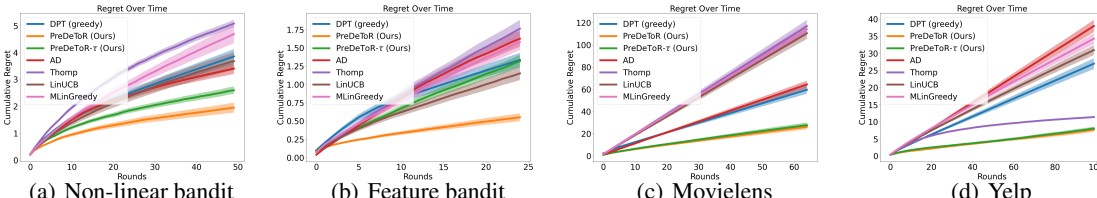

(a) Non-linear bandit     (b) Feature bandit     (c) Movielens     (d) Yelp

Figure 1: Non-linear regime. The horizontal axis is the number of rounds. Confidence bars show one standard error.

## 5   EMPIRICAL STUDY: LINEAR STRUCTURE AND UNDERSTANDING THE EXPLORATION OF PREDETOR

The previous experiments were conducted in a non-linear structured setting where we are unaware of a provably near-optimal algorithm. To assess how close PreDeToR's regret is to optimal, in this section, we consider a *linear* setting for which there exist well-understood algorithms (Abbasi-Yadkori et al., 2011; Lattimore & Szepesvári, 2020). Such algorithms provide a strong upper bound for PreDeToR. We summarize the key finding below:

**Finding 2:** PreDeToR (-$\tau$) matches the performance of the optimal algorithm LinUCB in linear bandit setting as it learns to exploit the latent structure across tasks from in-context data and without access to features.

In Figure 2 we first show the linear bandit setting for horizon $n = 25$, $M_{\text{pre}} = 200000$, $M_{\text{test}} = 200$, $A = 10$, and $d = 2$. Note that the length of the context (the number of rounds) is an artifact of the transformer architecture and computational complexity. This is because the self-attention takes in as input a length-$n$ sequence of tokens of size $d$, and requires $O\left(dn^2\right)$ time to compute the output (Keles et al., 2023). Further empirical setting details are stated in Appendix A.2.

We observe from Figure 2 that PreDeToR (-$\tau$) has lower cumulative regret than DPT-greedy, and AD. Note that for this low data (short horizon) regime, the DPT-greedy does not have a good estimation of $\widehat{a}_{m,*}$ which results in a poor prediction of optimal action $\widehat{a}_{m,t,*}$. This results in higher regret. Observe that PreDeToR (-$\tau$) performs quite similarly to LinUCB and lowers regret compared to Thomp which also shows that PreDeToR is able to exploit the latent linear structure and reward correlation of the underlying tasks. Note that LinUCB is close to the optimal algorithm for this linear bandit setting. PreDeToR outperforms AD as the main objective of AD is to match the performance of its demonstrator. In this short horizon, we see that MLinGreedy performs similarly to LinUCB.

We also show how the prediction error of the optimal action by PreDeToR is small compared to LinUCB in the linear bandit setting. In Figure 2(b) we first show how the 10 actions are distributed in the $M_{\text{test}} = 200$ test tasks. In Figure 2(b) for each bar, the frequency indicates the number of tasks where the action (shown in the x-axis) is the optimal action. Then, in Figure 2(c), we show the prediction error of PreDeToR (-$\tau$) for each task $m \in [M_{\text{test}}]$. The prediction error is calculated as $(\widehat{\mu}_{m,n,*}(a) - \mu_{m,*}(a))^2$ where $\widehat{\mu}_{m,n,*}(a) = \max_a \widehat{\boldsymbol{\theta}}_{m,n}^\top \mathbf{x}_m(a)$ is the empirical mean at the end of round $n$, and $\mu_{*,m}(a) = \max_a \boldsymbol{\theta}_{m,*}^\top \mathbf{x}_m(a)$ is the true mean of the optimal action in task $m$. Then we average the prediction error for the action $a \in [A]$ by the number of times the action $a$ is the optimal action in some task $m$. From the Figure 2(c), we see that for actions $\{2, 3, 5, 6, 7, 10\}$, the prediction error of PreDeToR is either close or smaller than LinUCB. Note that LinUCB estimates the empirical

mean directly from the test task, whereas PreDeToR has a strong prior based on the training data. So PreDeToR is able to estimate the reward of the optimal action quite well from the training dataset $\mathcal{D}_{\text{pre}}$. This shows the power of PreDeToR to go beyond the in-context decision-making setting studied in Lee et al. (2023); Lin et al. (2023); Ma et al. (2023); Sinii et al. (2023); Liu et al. (2023c) which require long horizons/trajectories and optimal action during training to learn a near-optimal policy. We discuss how exploration of PreDeToR ($-\tau$) results in low cumulative regret in Appendix A.11.

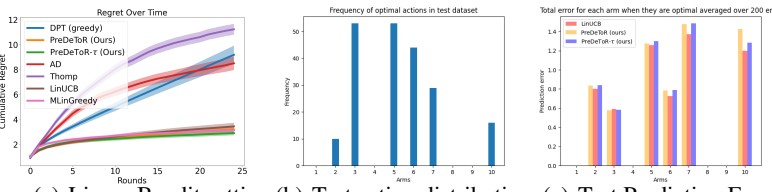

(a) Linear Bandit setting (b) Test action distribution (c) Test Prediction Error

Figure 2: Linear Expt. The horizontal axis is the number of rounds. Confidence bars show one standard error.

## 6 EMPIRICAL STUDY: IMPORTANCE OF SHARED STRUCTURE AND INTRODUCING NEW ARMS

One of our central claims is that PreDeToR ($-\tau$) internally learns and leverages the shared structure across the training and testing tasks. To validate this claim, in this section, we consider the introduction of new actions at test time that do *not* follow the structure of training time. These experiments are particularly important as they show the extent to which PreDeToR($-\tau$) is leveraging the latent structure and the shared correlation between the actions and rewards.

**Invariant actions:** We denote the set of actions fixed across the different tasks in the pretraining in-context dataset as $\mathcal{A}^{\text{inv}}$. Therefore these action features $\mathbf{x}(a) \in \mathbb{R}^d$ for $a \in \mathcal{A}^{\text{inv}}$ are fixed across the different tasks $m$. Note that these invariant actions help the transformer $\text{TF}_{\mathbf{w}}$ to learn the latent structure and the reward correlation across the different tasks. Therefore, as the structure breaks down, PreDeToR starts performing worse than other baselines.

**New actions:** However, we also want to test how robust is PreDeToR ($-\tau$) to new actions not seen during training time. To this effect, for each task $m \in [M_{\text{pre}}]$ and $m \in [M_{\text{test}}]$ we introduce $A - |\mathcal{A}^{\text{inv}}|$ new actions. *That is both for train and test tasks, we introduce new actions.* For each of these new actions $a \in [A - |\mathcal{A}^{\text{inv}}|]$ we choose the features $\mathbf{x}(m, a)$ randomly from $\mathcal{X} \subseteq \mathbb{R}^d$. Note the transformer now trains on a dataset $\mathcal{H}_m \subseteq \mathcal{D}_{\text{pre}} \neq \mathcal{D}_{\text{test}}$.

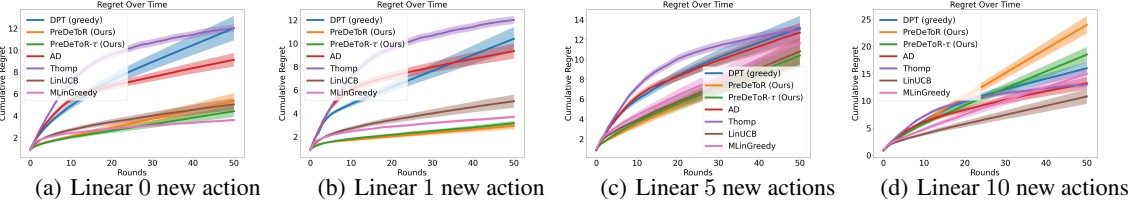

(a) Linear 0 new action (b) Linear 1 new action (c) Linear 5 new actions (d) Linear 10 new actions

Figure 3: New action experiments. The horizontal axis is the number of rounds. Confidence bars show one standard error.

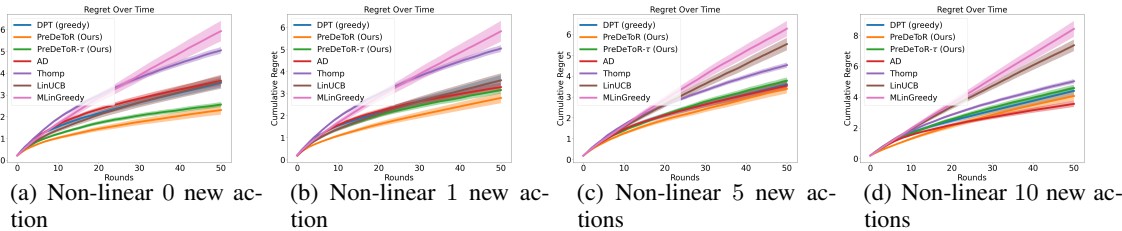

(a) Non-linear 0 new action (b) Non-linear 1 new action (c) Non-linear 5 new actions (d) Non-linear 10 new actions

Figure 4: New action experiments with non-linear setting.

**Baselines:** We implement the same baselines discussed in Section 4.

**Outcomes:** Again before presenting the result we discuss the main outcomes from our experimental results of introducing new actions during data collection and evaluation:

> **Finding 3:** PreDeToR (-$\tau$) performance degrades as the shared structure breaks down.

**Experimental Result:** We observe these outcomes in Figure 3 and Figure 4. We consider the linear and non-linear bandit setting of horizon $n = 50$, $M_{\text{pre}} = 100000$, $M_{\text{test}} = 200$, $A = 10$, and $d = 2$. Here during data collection and during collecting the test data, we randomly select between $0, 1, 5$, and $10$ new actions from $\mathbb{R}^d$ for each task $m$. So the number of invariant actions is $|\mathcal{A}^{\text{inv}}| \in \{10, 5, 1, 0\}$. Again, the demonstrator $\pi^w$ is the Thomp algorithm. From Figure 3(a), 3(b), 3(c), and 3(d), we observe that when the number of invariant actions is less than PreDeToR (-$\tau$) has lower cumulative regret than DPT-greedy, and AD. Observe that PreDeToR (-$\tau$) matches LinUCB and has lower regret than DPT-greedy, and AD when $\mathcal{A}^{\text{inv}}| \in \{10, 5, 1\}$. This shows that PreDeToR (-$\tau$) is able to exploit the latent linear structure of the underlying tasks. However, as the number of invariant actions decreases we see that PreDeToR(-$\tau$) performance drops and becomes similar to the unstructured bandits Thomp.

Similarly in Figure 4(a), 4(b), 4(c), and 4(d) we show the performance of PreDeToR in the non-linear bandit setting. Observe that LinUCB, MLinGreedy fails to perform well in this non-linear setting due to their assumption of linear rewards. Again note that PreDeToR (-$\tau$) has lower regret than DPT-greedy, and AD when $\mathcal{A}^{\text{inv}}| \in \{10, 1\}$. This shows that PreDeToR (-$\tau$) is able to exploit the latent linear structure of the underlying tasks. However, as the number of invariant actions decreases we see that PreDeToR(-$\tau$) performance drops and becomes similar to AD.

We also empirically study the test performance of PreDeToR (-$\tau$) in other *non-linear* bandit settings such as bilinear bandits (Appendix A.3), latent bandits (Appendix A.4), draw a connection between PreDeToR and Bayesian estimators (Appendix A.5), and perform sensitivity and ablation studies in Appendix A.6, A.8, A.9, A.10. We discuss data collection algorithms in Appendix A.13 and the offline setting in Appendix A.15. Due to space constraints, we refer the interested reader to the relevant section in the appendices.

## 7 THEORETICAL ANALYSIS OF GENERALIZATION

In this section, we present a theoretical analysis of how PreDeToR-$\tau$ generalizes to an unknown target task given a set of source tasks. We observe that PreDeToR-$\tau$'s performance hinges on a low excess error on the predicted reward of the actions of the unknown target task based on the in-context data. Thus, in our analysis, we show that, in low-data regimes, PreDeToR-$\tau$ has a low expected excess risk for the unknown target task as the number of source tasks increases. This is summarized as follows:

> **Finding 4:** PreDeToR (-$\tau$) has a low expected excess risk for the unknown target task as the number of source tasks increases. Moreover, the transfer learning risk of PreDeToR-$\tau$ (once trained on the $M$ source tasks) scales with $\widetilde{O}(1/\sqrt{M})$.

To show this, we proceed as follows: Suppose we have the training data set $\mathcal{H}_{\text{all}} = \{\mathcal{H}_m\}_{m=1}^{M_{\text{pre}}}$, where the task $m \sim \mathcal{T}$ with a distribution $\mathcal{T}$ and the task data $\mathcal{H}_m$ is generated from a distribution $\mathcal{D}_{\text{pre}}(\cdot|m)$. For illustration purposes, here we consider the training data distribution $\mathcal{D}_{\text{pre}}(\cdot|m)$ where the actions are sampled following soft-LinUCB (a stochastic variant of LinUCB) (Chu et al., 2011). Given the loss function in Equation (3), we can define the task $m$ training loss of PreDeToR-$\tau$ as $\widehat{\mathcal{L}}_m(\text{TF}^{\mathbf{r}}_{\boldsymbol{\Theta}}) = \frac{1}{n} \sum_{t=1}^{n} \ell(r_{m,t}, \text{TF}^{\mathbf{r}}_{\boldsymbol{\Theta}}(\widehat{r}_{m,t}(I_{m,t})|\mathcal{H}_m^t)) = \frac{1}{n} \sum_{t=1}^{n} (\text{TF}^{\mathbf{r}}_{\boldsymbol{\Theta}}(\widehat{r}_{m,t}(I_{m,t})|\mathcal{H}_m^t) - r_{m,t})^2$. We drop the notation $\boldsymbol{\Theta}, \mathbf{r}$ from $\text{TF}^{\mathbf{r}}_{\boldsymbol{\Theta}}$ for simplicity and let $M = M_{\text{pre}}$. We define

$$\widehat{\text{TF}} = \underset{\text{TF} \in \text{Alg}}{\arg\min} \widehat{\mathcal{L}}_{\mathcal{H}_{\text{all}}}(\text{TF}) := \frac{1}{M} \sum_{m=1}^{M} \widehat{\mathcal{L}}_m(\text{TF}), \quad (\text{ERM}) \qquad (4)$$

where Alg denotes the space of algorithms induced by the TF. Let $\mathcal{L}_m(\text{TF}) = \mathbb{E}_{\mathcal{H}_m}[\widehat{\mathcal{L}}_m(\text{TF})]$ and $\mathcal{L}_{\text{MTL}}(\text{TF}) = \mathbb{E}[\widehat{\mathcal{L}}_{\mathcal{H}_{\text{all}}}(\text{TF})] = \frac{1}{M} \sum_{m=1}^{M} \mathcal{L}_m(\text{TF})$ be the corresponding population risks. For

the ERM in equation 4, we want to bound the following excess Multi-Task Learning (MTL) risk of PreDeToR-$\tau$

$$\mathcal{R}_{\text{MTL}}(\widehat{\text{TF}}) = \mathcal{L}_{\text{MTL}}(\widehat{\text{TF}}) - \min_{\text{TF} \in \text{Alg}} \mathcal{L}_{\text{MTL}}(\text{TF}). \tag{5}$$

Note that for in-context learning, a training sample $(I_t, r_t)$ impacts all future decisions of the algorithm from time step $t+1$ to $n$. Therefore, we need to control the stability of the input perturbation of the learning algorithm learned by the transformer. We introduce the following stability condition.

**Assumption 7.1.** (Error stability (Bousquet & Elisseeff, 2002; Li et al., 2023)). Let $\mathcal{H} = (I_t, r_t)_{t=1}^n$ be a sequence in $[A] \times [0, 1]$ with $n \geq 1$ and $\mathcal{H}'$ be the sequence where the $t'$th sample of $\mathcal{H}$ is replaced by $(I'_t, r'_t)$. Error stability holds for a distribution $(I, r) \sim \mathcal{D}$ if there exists a $K > 0$ such that for any $\mathcal{H}, (I'_t, r'_t) \in ([A] \times [0, 1]), t \leq n$, and $\text{TF} \in \text{Alg}$, we have

$$\left| \mathbb{E}_{(I,r)} \left[ \ell(r, \text{TF}(\widehat{r}(I)|\mathcal{H})) - \ell\left(r, \text{TF}(\widehat{r}(I)|\mathcal{H}')\right) \right] \right| \leq \frac{K}{n}.$$

Let $\rho$ be a distance metric on Alg. Pairwise error stability holds if for all $\text{TF}, \text{TF}' \in \text{Alg}$ we have

$$\left| \mathbb{E}_{(x,y)} \left[ \ell(r, \text{TF}(\widehat{r}(I)|\mathcal{H})) - \ell\left(r, \text{TF}'(\widehat{r}(I)|\mathcal{H})\right) - \ell(r, \text{TF}(\widehat{r}(I)|\mathcal{H}')) + \ell\left(r, \text{TF}'(\widehat{r}(I)|\mathcal{H}')\right) \right] \right| \leq \frac{K\rho(\text{TF}, \text{TF}')}{n}.$$

Now we present the Multi-task learning (MTL) risk of PreDeToR-$\tau$.

**Theorem 7.2.** *(**PreDeToR risk**) Suppose error stability Assumption 7.1 holds and assume loss function $\ell(\cdot, \cdot)$ is $C$-Lipschitz for all $r_t \in [0, B]$ and horizon $n \geq 1$. Let $\widehat{\text{TF}}$ be the empirical solution of (ERM) and $\mathcal{N}(\mathcal{A}, \rho, \epsilon)$ be the covering number of the algorithm space Alg following Definition C.2 and C.3. Then with probability at least $1 - 2\delta$, the excess MTL risk of PreDeToR-$\tau$ is bounded by*

$$\mathcal{R}_{\text{MTL}}(\widehat{\text{TF}}) \leq 4\frac{C}{\sqrt{nM}} + 2(B + K \log n)\sqrt{\frac{\log(\mathcal{N}(\text{Alg}, \rho, \varepsilon)/\delta)}{cnM}},$$

*where $\mathcal{N}(\text{Alg}, \rho, \varepsilon)$ is the covering number of transformer $\widehat{\text{TF}}$ and $\epsilon = 1/\sqrt{nM}$.*

The proof of Theorem 7.2 is provided in Appendix C.1. From Theorem 7.2 we see that in low-data regime with a small horizon $n$, as the number of tasks $M$ increases the MTL risk decreases. We further discuss the stability factor $K$ and covering number $\mathcal{N}(\text{Alg}, \rho, \varepsilon)$ in Remark C.4, and C.5.

We now present the transfer learning risk of PreDeToR-$\tau$ for an unknown target task $g \sim \mathcal{T}$ with the test dataset $\mathcal{H}_g \sim \mathcal{D}_{\text{test}}(\cdot|g)$. Note that the test data distribution $\mathcal{D}_{\text{test}}(\cdot|g)$ is such that the actions are sampled following soft-LinUCB.

**Theorem 7.3.** *(**Transfer risk**) Consider the setting of Theorem 7.2 and assume the training source tasks are independently drawn from task distribution $\mathcal{T}$. Let $\widehat{\text{TF}}$ be the empirical solution of (ERM) and $g \sim \mathcal{T}$. Define the expected excess transfer learning risk $\mathbb{E}_g[\mathcal{R}_g] = \mathbb{E}_g\left[\mathcal{L}_g(\widehat{\text{TF}})\right] - \arg\min_{\text{TF} \in \text{Alg}} \mathbb{E}_g\left[\mathcal{L}_g(\text{TF})\right]$. Then with probability at least $1 - 2\delta$, the $\mathbb{E}_g\left[\mathcal{R}_g\right] \leq 4\frac{C}{\sqrt{M}} + 2B\sqrt{\frac{\log(\mathcal{N}(\text{Alg}, \rho, \varepsilon)/\delta)}{M}}$, where $\mathcal{N}(\text{Alg}, \rho, \varepsilon)$ is the covering number of $\widehat{\text{TF}}$ and $\epsilon = \frac{1}{\sqrt{M}}$.*

The proof is given in Appendix C.2. This shows that for the transfer learning risk of PreDeToR-$\tau$ (once trained on the $M$ source tasks) scales with $\widetilde{O}(1/\sqrt{M})$. This is because the unseen target task $g \sim \mathcal{T}$ induces a distribution shift, which, typically, cannot be mitigated with more samples $n$ per task. A similar observation is provided in Lin et al. (2023). We further discuss this in Remark C.7. We also observe a similar phenomenon empirically; see the discussion in Appendix A.14.

## 8 CONCLUSIONS, LIMITATIONS AND FUTURE WORKS

In this paper, we studied the supervised pretraining of decision transformers in the multi-task structured bandit setting when the knowledge of the optimal action is unavailable. Moreover, our proposed methods PreDeToR (-$\tau$) do not need to know the action representations or the reward structure and learn these in-context with the help of offline data. The PreDeToR (-$\tau$) predict the reward for the next action of each action during pretraining and can generalize well in-context in several regimes spanning low-data, new actions, and structured bandit settings like linear, non-linear, bilinear, latent bandits. The PreDeToR (-$\tau$) outperforms other in-context algorithms like AD, DPT-greedy in most of the experiments. Finally, we theoretically analyze PreDeToR-$\tau$ and show that pretraining it in $M$ source tasks leads to a low expected excess error on a target task drawn from the same task distribution $\mathcal{T}$. In future, we want to extend our PreDeToR (-$\tau$) to MDP setting (Sutton & Barto, 2018; Agarwal et al., 2019), and constraint MDP setting (Efroni et al., 2020; Gu et al., 2022).

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

# A  APPENDIX

## A.1  RELATED WORKS

In this section, we briefly discuss related works.

In-context decision making (Laskin et al., 2022; Lee et al., 2023) has emerged as an attractive alternative in Reinforcement Learning (RL) compared to updating the model parameters after collection of new data (Mnih et al., 2013; François-Lavet et al., 2018). In RL the contextual data takes the form of state-action-reward tuples representing a dataset of interactions with an unknown environment (task). In this paper, we will refer to this as the in-context data. Recall that in many real-world settings, the underlying task can be structured with correlated features, and the reward can be highly non-linear. So specialized bandit algorithms fail to learn in these tasks. To circumvent this issue, a learner can first collect in-context data consisting of just action indices $I_t$ and rewards $r_t$. Then it can leverage the representation learning capability of deep neural networks to learn a pattern across the in-context data and subsequently derive a near-optimal policy (Lee et al., 2023; Mirchandani et al., 2023). We refer to this learning framework as an in-context decision-making setting.

The in-context decision-making setting of Sinii et al. (2023) also allows changing the action space by learning an embedding over the action space yet also requires the optimal action during training. In contrast we do not require the optimal action as well as show that we can generalize to new actions without learning an embedding over them. Similarly, Lin et al. (2023) study the in-context decision-making setting of Laskin et al. (2022); Lee et al. (2023), but they also require a greedy approximation of the optimal action. The Ma et al. (2023) also studies a similar setting for hierarchical RL where they stitch together sub-optimal trajectories and predict the next action during test time. Similarly, Liu et al. (2023c) studies the in-context decision-making setting to predict action instead of learning a reward correlation from a short horizon setting. In contrast we do not require a greedy approximation of the optimal action, deal with short horizon setting and changing action sets during training and testing, and predict the estimated means of the actions instead of predicting the optimal action. A survey of the in-context decision-making approaches can be found in Liu et al. (2023a).

In the in-context decision-making setting, the learning model is first trained on supervised input-output examples with the in-context data during training. Then during test time, the model is asked to complete a new input (related to the context provided) without any update to the model parameters (Xie et al., 2021; Min et al., 2022). Motivated by this, Lee et al. (2023) recently proposed the Decision Pretrained Transformers (DPT) that exhibit the following properties: **(1)** During supervised pretraining of DPT, predicting optimal actions alone gives rise to near-optimal decision-making algorithms for unforeseen task during test time. Note that DPT does not update model parameters during test time and, therefore, conducts in-context learning on the unforeseen task. **(2)** DPT improves over the in-context data used to pretrain it by exploiting latent structure. However, DPT either requires the optimal action during training or if it needs to approximate the optimal action. For approximating the optimal action, it requires a large amount of data from the underlying task.

At the same time, learning the underlying data pattern from a few examples during training is becoming more relevant in many domains like chatbot interaction (Madotto et al., 2021; Semnani et al., 2023), recommendation systems, healthcare (Ge et al., 2022; Liu et al., 2023b), etc. This is referred to as few-shot learning. However, most current RL decision-making systems (including in-context learners like DPT) require an enormous amount of data to learn a good policy.

The in-context learning framework is related to the meta-learning framework (Bengio et al., 1990; Schaul & Schmidhuber, 2010). Broadly, these techniques aim to learn the underlying latent shared structure within the training distribution of tasks, facilitating faster learning of novel tasks during test time. In the context of decision-making and reinforcement learning (RL), there exists a frequent choice regarding the specific 'structure' to be learned, be it the task dynamics (Fu et al., 2016; Nagabandi et al., 2018; Landolfi et al., 2019), a task context identifier (Rakelly et al., 2019; Zintgraf et al., 2019; Liu et al., 2021), or temporally extended skills and options (Perkins & Precup, 1999; Gupta et al., 2018; Jiang et al., 2022).

However, as we noted in the Section 1, one can do a greedy approximation of the optimal action from the historical data using a weak demonstrator and a neural network policy (Finn et al., 2017; Rothfuss et al., 2018). Moreover, the in-context framework generally is more agnostic where it learns

the policy of the demonstrator (Duan et al., 2016; Wang et al., 2016; Mishra et al., 2017). Note that both DPT-greedy and PreDeToR are different than algorithmic distillation (Laskin et al., 2022; Lu et al., 2023) as they do not distill an existing RL algorithm. moreover, in contrast to DPT-greedy which is trained to predict the optimal action, the PreDeToR is trained to predict the reward for each of the actions. This enables the PreDeToR (similar to DPT-greedy) to show to potentially emergent online and offline strategies at test time that automatically align with the task structure, resembling posterior sampling.

As we discussed in the Section 1, in decision-making, RL, and imitation learning the transformer models are trained using autoregressive action prediction (Yang et al., 2023). Similar methods have also been used in Large language models (Vaswani et al., 2017; Roberts et al., 2019). One of the more notable examples is the Decision Transformers (abbreviated as DT) which utilizes a transformer to autoregressively model sequences of actions from offline experience data, conditioned on the achieved return (Chen et al., 2021; Janner et al., 2021). This approach has also been shown to be effective for multi-task settings (Lee et al., 2022), and multi-task imitation learning with transformers (Reed et al., 2022; Brohan et al., 2022; Shafiullah et al., 2022). However, the DT methods are not known to improve upon their in-context data, which is the main thrust of this paper (Brandfonbrener et al., 2022; Yang et al., 2022b).

Our work is also closely related to the offline RL setting. In offline RL, the algorithms can formulate a policy from existing data sets of state, action, reward, and next-state interactions. Recently, the idea of pessimism has also been introduced in an offline setting to address the challenge of distribution shift (Kumar et al., 2020; Yu et al., 2021; Liu et al., 2020; Ghasemipour et al., 2022). Another approach to solve this issue is policy regularization (Fujimoto et al., 2019; Kumar et al., 2019; Wu et al., 2019; Siegel et al., 2020; Liu et al., 2019), or reuse data for related task (Li et al., 2020; Mitchell et al., 2021), or additional collection of data along with offline data (Pong et al., 2022). However, all of these approaches still have to take into account the issue of distributional shifts. In contrast PreDeToR and DPT-greedy leverages the decision transformers to avoid these issues. Both of these methods can also be linked to posterior sampling. Such connections between sequence modeling with transformers and posterior sampling have also been made in Chen et al. (2021); Müller et al. (2021); Lee et al. (2023); Yang et al. (2023).

## A.2 EXPERIMENTAL SETTING INFORMATION AND DETAILS OF BASELINES

In this section, we describe in detail the experimental settings and some baselines.

### A.2.1 EXPERIMENTAL DETAILS

**Linear Bandit:** We consider the setting when $f(\mathbf{x}, \boldsymbol{\theta}_*) = \mathbf{x}^\top \boldsymbol{\theta}_*$. Here $\mathbf{x} \in \mathbb{R}^d$ is the action feature and $\boldsymbol{\theta}_* \in \mathbb{R}^d$ is the hidden parameter. For every experiment, we first generate tasks from $\mathcal{T}_{\text{pre}}$. Then we sample a fixed set of actions from $\mathcal{N}(\mathbf{0}, \mathbf{I}_d/d)$ in $\mathbb{R}^d$ and this constitutes the features. Then for each task $m \in [M]$ we sample $\boldsymbol{\theta}_{m,*} \sim \mathcal{N}(\mathbf{0}, \mathbf{I}_d/d)$ to produce the means $\mu(m, a) = \langle \boldsymbol{\theta}_{m,*}, \mathbf{x}(m, a) \rangle$ for $a \in \mathcal{A}$ and $m \in [M]$. Finally, note that we do not shuffle the data as the order matters. Also in this setting $\mathbf{x}(m, a)$ for each $a \in \mathcal{A}$ is fixed for all tasks $m$.

**Non-Linear Bandit:** We now consider the setting when $f(\mathbf{x}, \boldsymbol{\theta}_*) = 1/(1 + 0.5 \cdot \exp(2 \cdot \exp(-\mathbf{x}^\top \boldsymbol{\theta}_*)))$. Again, here $\mathbf{x} \in \mathbb{R}^d$ is the action feature, and $\boldsymbol{\theta}_* \in \mathbb{R}^d$ is the hidden parameter. Note that this is different than the generalized linear bandit setting (Filippi et al., 2010; Li et al., 2017). Again for every experiment, we first generate tasks from $\mathcal{T}_{\text{pre}}$. Then we sample a fixed set of actions from $\mathcal{N}(\mathbf{0}, \mathbf{I}_d/d)$ in $\mathbb{R}^d$ and this constitutes the features. Then for each task $m \in [M]$ we sample $\boldsymbol{\theta}_{m,*} \sim \mathcal{N}(\mathbf{0}, \mathbf{I}_d/d)$ to produce the means $\mu(m, a) = 1/(1 + 0.5 \cdot \exp(2 \cdot \exp(-\mathbf{x}(m, a)^\top \boldsymbol{\theta}_{m,*})))$ for $a \in \mathcal{A}$ and $m \in [M]$. Again note that in this setting $\mathbf{x}(m, a)$ for each $a \in \mathcal{A}$ is fixed for all tasks $m$.

We use NVIDIA GeForce RTX 3090 GPU with 24GB RAM to load the GPT 2 Large Language Model. This requires less than 2GB RAM without data, and with large context may require as much as 20GB RAM.

### A.2.2 DETAILS OF BASELINES

**(1) Thomp:** This baseline is the stochastic $A$-action bandit Thompson Sampling algorithm from Thompson (1933); Agrawal & Goyal (2012); Russo et al. (2018); Zhu & Tan (2020). We briefly describe the algorithm below: At every round $t$ and each action $a$, Thomp samples $\gamma_{m,t}(a) \sim \mathcal{N}(\widehat{\mu}_{m,t-1}(a), \sigma^2/N_{m,t-1}(a))$, where $N_{m,t-1}(a)$ is the number of times the action $a$ has been selected till $t-1$, and $\widehat{\mu}_{m,t-1}(a) = \frac{\sum_{s=1}^{t-1}\widehat{r}_{m,s}\mathbf{1}(I_s=a)}{N_{m,t-1}(a)}$ is the empirical mean. Then the action selected at round $t$ is $I_t = \arg\max_a \gamma_{m,t}(a)$. Observe that Thomp is not a deterministic algorithm like UCB (Auer et al., 2002). So we choose Thomp as the weak demonstrator $\pi^w$ because it is more exploratory than UCB and also chooses the optimal action, $a_{m,*}$, a sufficiently large number of times. Thomp is a weak demonstrator as it does not have access to the feature set $\mathcal{X}$ for any task $m$.

**(2) LinUCB:** (Linear Upper Confidence Bound): This baseline is the Upper Confidence Bound algorithm for the linear bandit setting that selects the action $I_t$ at round $t$ for task $m$ that is most optimistic and reduces the uncertainty of the task unknown parameter $\boldsymbol{\theta}_{m,*}$. To balance exploitation and exploration between choosing different items the LinUCB computes an upper confidence value to the estimated mean of each action $\mathbf{x}_{m,a} \in \mathcal{X}$. This is done as follows: At every round $t$ for task $m$, it calculates the ucb value $B_{m,a,t}$ for each action $\mathbf{x}_{m,a} \in \mathcal{X}$ such that $B_{m,a,t} = \mathbf{x}_{m,a}^\top \widehat{\boldsymbol{\theta}}_{m,t-1} + \alpha\|\mathbf{x}_{m,a}\|_{\boldsymbol{\Sigma}_{m,t-1}^{-1}}$ where $\alpha > 0$ is a constant and $\widehat{\boldsymbol{\theta}}_{m,t}$ is the estimate of the model parameter $\boldsymbol{\theta}_{m,*}$ at round $t$. Here, $\boldsymbol{\Sigma}_{m,t-1} = \sum_{s=1}^{t-1} \mathbf{x}_{m,s}\mathbf{x}_{m,s}^\top + \lambda\mathbf{I}_d$ is the data covariance matrix or the arms already tried. Then it chooses $I_t = \arg\max_a B_{m,a,t}$. Note that LinUCB is a *strong* demonstrator that we give oracle access to the features of each action; other algorithms do not observe the features. Hence, in linear bandits, LinUCB provides an approximate upper bound on the performance of all algorithms.

**(3) MLinGreedy:** This is the multi-task linear regression bandit algorithm proposed by Yang et al. (2021). This algorithm assumes that there is a common low dimensional feature extractor $\mathbf{B} \in \mathbb{R}^{k \times d}$, $k \leq d$ shared between the tasks and the rewards per task $m$ are linearly dependent on a hidden parameter $\boldsymbol{\theta}_{m,*}$. Under a diversity assumption (which may not be satisfied in real data) and $\mathbf{W} = [\mathbf{w}_1, \ldots, \mathbf{w}_M]$ they assume $\boldsymbol{\Theta} = [\boldsymbol{\theta}_{1,*}, \ldots, \boldsymbol{\theta}_{M,*}] = \mathbf{B}\mathbf{W}$. During evaluation MLinGreedy estimates the $\widehat{\mathbf{B}}$ and $\widehat{\mathbf{W}}$ from training data and fit $\widehat{\boldsymbol{\theta}}_m = \widehat{\mathbf{B}}\widehat{\mathbf{w}}_m$ per task and selects action greedily based on $I_{m,t} = \arg\max_a \mathbf{x}_{m,a}^\top \widehat{\boldsymbol{\theta}}_{m,*}$. Finally, note that MLinGreedy requires access to the action features to estimate $\widehat{\boldsymbol{\theta}}_m$ and select actions as opposed to DPT, AD, and PreDeToR.

### A.3 EMPIRICAL STUDY: BILINEAR BANDITS

In this section, we discuss the performance of PreDeToR against the other baselines in the bilinear setting. Again note that the number of tasks $M_{\text{pre}} \gg A \geq n$. Through this experiment, we want to evaluate the performance of PreDeToR to exploit the underlying latent structure and reward correlation when the horizon is small, the number of tasks is large, and understand its performance in the bilinear bandit setting (Jun et al., 2019; Lu et al., 2021; Kang et al., 2022; Mukherjee et al., 2023). Note that this setting also goes beyond the linear feedback model (Abbasi-Yadkori et al., 2011; Lattimore & Szepesvári, 2020) and is related to matrix bandits (Yang & Wang, 2020).

**Bilinear bandit setting:** In the bilinear bandits the learner is provided with two sets of action sets, $\mathcal{X} \subseteq \mathbb{R}^{d_1}$ and $\mathcal{Z} \subseteq \mathbb{R}^{d_2}$ which are referred to as the left and right action sets. At every round $t$ the learner chooses a pair of actions $\mathbf{x}_t \in \mathcal{X}$ and $\mathbf{z}_t \in \mathcal{Z}$ and observes a reward

$$r_t = \mathbf{x}_t^\top \boldsymbol{\Theta}_* \mathbf{z}_t + \eta_t$$

where $\boldsymbol{\Theta}_* \in \mathbb{R}^{d_1 \times d_2}$ is the unknown hidden matrix which is also low-rank. The $\eta_t$ is a $\sigma^2$ sub-Gaussian noise. In the multi-task bilinear bandit setting we now have a set of $M$ tasks where the reward for the $m$-th task at round $t$ is given by

$$r_{m,t} = \mathbf{x}_{m,t}^\top \boldsymbol{\Theta}_{m,*} \mathbf{z}_{m,t} + \eta_{m,t}.$$

Here $\boldsymbol{\Theta}_{m,*} \in \mathbb{R}^{d_1 \times d_2}$ is the unknown hidden matrix for each task $m$, which is also low-rank. The $\eta_{m,t}$ is a $\sigma^2$ sub-Gaussian noise. Let $\kappa$ be the rank of each of these matrices $\boldsymbol{\Theta}_{m,*}$.

A special case is the rank 1 structure where $\boldsymbol{\Theta}_{m,*} = \boldsymbol{\theta}_{m,*}\boldsymbol{\theta}_{m,*}^{\top}$ where $\boldsymbol{\Theta}_{m,*} \in \mathbb{R}^{d \times d}$ and $\boldsymbol{\theta}_{m,*} \in \mathbb{R}^{d}$ for each task $m$. Let the left and right action sets be also same such that $\mathbf{x}_{m,t} \in \mathcal{X} \subseteq \mathbb{R}^{d}$. Observe then that the reward for the $m$-th task at round $t$ is given by

$$r_{m,t} = \mathbf{x}_{m,t}^{\top}\boldsymbol{\Theta}_{m,*}\mathbf{x}_{m,t} + \eta_{m,t} = (\mathbf{x}_{m,t}^{\top}\boldsymbol{\theta}_{m,*})^2 + \eta_{m,t}.$$

This special case is studied in Chaudhuri et al. (2017).

**Baselines:** We again implement the same baselines discussed in Section 4. The baselines are PreDeToR, PreDeToR-$\tau$, DPT-greedy, and Thomp. Note that we do not implement the LinUCB and MLinGreedy for the bilinear bandit setting. However, we now implement the LowOFUL (Jun et al., 2019) which is optimal in the bilinear bandit setting.

**LowOFUL:** The LowOFUL algorithm first estimates the unknown parameter $\boldsymbol{\Theta}_{m,*}$ for each task $m$ using E-optimal design (Pukelsheim, 2006; Fedorov, 2013; Jun et al., 2019) for $n_1$ rounds. Let $\widehat{\boldsymbol{\Theta}}_{m,n_1}$ be the estimate of $\boldsymbol{\Theta}_{m,*}$ at the end of $n_1$ rounds. Let the SVD of $\widehat{\boldsymbol{\Theta}}_{m,n_1}$ be given by $\mathrm{SVD}(\widehat{\boldsymbol{\Theta}}_{m,n_1}) = \widehat{\mathbf{U}}_{m,n_1}\widehat{\mathbf{S}}_{m,n_1}\widehat{\mathbf{V}}_{m,n_1}$. Then LowOFUL rotates the actions as follows:

$$\mathcal{X}'_m = \left\{ \left[\widehat{\mathbf{U}}_{m,n_1}\widehat{\mathbf{U}}_{m,n_1}^{\perp}\right]^{\top}\mathbf{x}_m : \mathbf{x}_m \in \mathcal{X} \right\} \text{ and } \mathcal{Z}' = \left\{ \left[\widehat{\mathbf{V}}_{m,n_1}\widehat{\mathbf{V}}_{m,n_1}^{\perp}\right]^{\top}\mathbf{z}_m : \mathbf{z}_m \in \mathcal{Z} \right\}.$$

Then defines a vectorized action set for each task $m$ so that the last $(d_1 - \kappa) \cdot (d_2 - \kappa)$ components are from the complementary subspaces:

$$\widetilde{\mathcal{A}}_m = \left\{ \left[\mathrm{vec}\left(\mathbf{x}_{m,1:\kappa}\mathbf{z}_{m,1:\kappa}^{\top}\right); \mathrm{vec}\left(\mathbf{x}_{m,\kappa+1:d_1}\mathbf{z}_{m,1:\kappa}^{\top}\right); \mathrm{vec}\left(\mathbf{x}_{m,1:\kappa}\mathbf{z}_{m,\kappa+1:d_2}^{\top}\right); \right.$$
$$\left. \mathrm{vec}\left(\mathbf{x}_{m,\kappa+1:d_1}\mathbf{z}_{m,\kappa+1:d_2}^{\top}\right)\right] \in \mathbb{R}^{d_1 d_2} : \mathbf{x}_m \in \mathcal{X}'_m, \mathbf{z}_m \in \mathcal{Z}'_m \right\}.$$

Finally for $n_2 = n - n_1$ rounds, LowOFUL invokes the specialized OFUL algorithm (Abbasi-Yadkori et al., 2011) for the rotated action set $\widetilde{\mathcal{A}}_m$ with the low dimension $k = (d_1 + d_2)\kappa - \kappa^2$. Note that the LowOFUL runs the per-task low dimensional OFUL algorithm rather than learning the underlying structure across the tasks (Mukherjee et al., 2023).

**Outcomes:** We first discuss the main outcomes of our experimental results for increasing the horizon:

> **Finding 5:** PreDeToR (-$\tau$) outperforms DPT-greedy, AD, and matches the performance of LowOFUL in bilinear bandit setting.

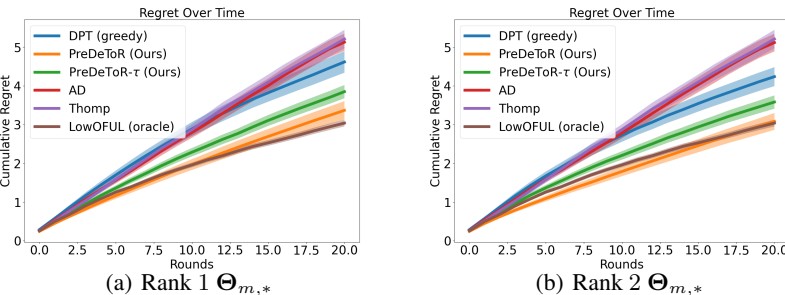

(a) Rank 1 $\boldsymbol{\Theta}_{m,*}$      (b) Rank 2 $\boldsymbol{\Theta}_{m,*}$

Figure 5: Experiment with bilinear bandits. The y-axis shows the cumulative regret.

**Experimental Result:** We observe these outcomes in Figure 5. In Figure 5(a) we experiment with rank 1 hidden parameter $\boldsymbol{\Theta}_{m,*}$ and set horizon $n = 20$, $M_{\mathrm{pre}} = 200000$, $M_{\mathrm{test}} = 200$, $A = 30$, and $d = 5$. In Figure 5(b) we experiment with rank 2 hidden parameter $\boldsymbol{\Theta}_{m,*}$ and set horizon $n = 20$, $M_{\mathrm{pre}} = 250000$, $M_{\mathrm{test}} = 200$, $A = 25$, and $d = 5$. Again, the demonstrator $\pi^w$ is the Thomp algorithm. We observe that PreDeToR has lower cumulative regret than DPT-greedy, AD and Thomp. Note that for any task $m$ for the horizon 20 the Thomp will be able to sample all the actions at most once. Note that for this small horizon setting the DPT-greedy does not have a good estimation of $\widehat{a}_{m,*}$ which results in a poor prediction of optimal action $\widehat{a}_{m,t,*}$. In contrast PreDeToR learns the

correlation of rewards across tasks and can perform well. Observe from Figure 5(a), and 5(b) that PreDeToR has lower regret than Thomp and matches LowOFUL. Also, in this low-data regime it is not enough for LowOFUL to learn the underlying $\boldsymbol{\Theta}_{m,*}$ with high precision. Hence, PreDeToR also has slightly lower regret than LowOFUL. Note that the main objective of AD is to match the performance of its demonstrator. Most importantly it shows that PreDeToR can exploit the underlying latent structure and reward correlation better than DPT-greedy, and AD.

## A.4 EMPIRICAL STUDY: LATENT BANDITS

In this section, we discuss the performance of PreDeToR ($-\tau$) against the other baselines in the latent bandit setting and create a generalized bilinear bandit setting. Note that the number of tasks $M_{\text{pre}} \gg A \geq n$. Using this experiment, we want to evaluate the ability of PreDeToR ($-\tau$) to exploit the underlying reward correlation when the horizon is small, the number of tasks is large, and understand its performance in the latent bandit setting (Hong et al., 2020; Maillard & Mannor, 2014; Pal et al., 2023; Kveton et al., 2017). We create a latent bandit setting which generalizes the bilinear bandit setting (Jun et al., 2019; Lu et al., 2021; Kang et al., 2022; Mukherjee et al., 2023). Again note that this setting also goes beyond the linear feedback model (Abbasi-Yadkori et al., 2011; Lattimore & Szepesvári, 2020) and is related to matrix bandits (Yang & Wang, 2020).

**Latent bandit setting:** In this special multi-task latent bandits the learner is again provided with two sets of action sets, $\mathcal{X} \subseteq \mathbb{R}^{d_1}$ and $\mathcal{Z} \subseteq \mathbb{R}^{d_2}$ which are referred to as the left and right action sets. The reward for the $m$-th task at round $t$ is given by

$$r_{m,t} = \mathbf{x}_{m,t}^\top \underbrace{\left(\boldsymbol{\Theta}_{m,*} + \mathbf{U}\mathbf{V}^\top\right)}_{\mathbf{z}_{m_*}} \mathbf{z}_{m,t} + \eta_{m,t}.$$

Here $\boldsymbol{\Theta}_{m,*} \in \mathbb{R}^{d_1 \times d_2}$ is the unknown hidden matrix for each task $m$, which is also low-rank. Additionally, all the tasks share a *common latent parameter matrix* $\mathbf{U}\mathbf{V}^\top \in \mathbb{R}^{d_1 \times d_2}$ which is also low rank. Hence the learner needs to learn the latent parameter across the tasks hence the name latent bandits. Finally, the $\eta_{m,t}$ is a $\sigma^2$ sub-Gaussian noise. Let $\kappa$ be the rank of each of these matrices $\boldsymbol{\Theta}_{m,*}$ and $\mathbf{U}\mathbf{V}^\top$. Again special case is the rank 1 structure where the reward for the $m$-th task at round $t$ is given by

$$r_{m,t} = \mathbf{x}_{m,t}^\top \underbrace{\left(\boldsymbol{\theta}_{m,*}\boldsymbol{\theta}_{m,*}^\top + \mathbf{u}\mathbf{v}^\top\right)}_{\mathbf{z}_{m,*}} \mathbf{x}_{m,t} + \eta_{m,t}.$$

where $\boldsymbol{\theta}_{m,*} \in \mathbb{R}^d$ for each task $m$ and $\mathbf{u}, \mathbf{v} \in \mathbb{R}^d$. Note that the left and right action sets are the same such that $\mathbf{x}_{m,t} \in \mathcal{X} \subseteq \mathbb{R}^d$.

**Baselines:** We again implement the same baselines discussed in Section 4. The baselines are PreDeToR, PreDeToR-$\tau$, DPT-greedy, AD, Thomp, and LowOFUL. However, we now implement a special LowOFUL (stated in Appendix A.3) which has knowledge of the shared latent parameters $\mathbf{U}$, and $\mathbf{V}$. We call this the LowOFUL (oracle) algorithm. Therefore LowOFUL (oracle) has knowledge of the problem parameters in the latent bandit setting and hence the name. Again note that we do not implement the LinUCB and MLinGreedy for the latent bandit setting.

**Outcomes:** We first discuss the main outcomes of our experimental results for increasing the horizon:

> **Finding 6:** PreDeToR ($-\tau$) outperforms DPT-greedy, AD, and matches the performance of LowOFUL (oracle) in latent bandit setting.

**Experimental Result:** We observe these outcomes in Figure 6. In Figure 6(a) we experiment with rank 1 hidden parameter $\boldsymbol{\theta}_{m,*}\boldsymbol{\theta}_{m,*}^\top$ and latent parameters $\mathbf{u}\mathbf{v}^\top$ shared across the tasks and set horizon $n = 20$, $M_{\text{pre}} = 200000$, $M_{\text{test}} = 200$, $A = 30$, and $d = 5$. In Figure 6(b) we experiment with rank 2 hidden parameter $\boldsymbol{\Theta}_{m,*}$, and latent parameters $\mathbf{U}\mathbf{V}^\top$ and set horizon $n = 20$, $M_{\text{pre}} = 250000$, $M_{\text{test}} = 200$, $A = 25$, and $d = 5$. In Figure 6(c) we experiment with rank 3 hidden parameter $\boldsymbol{\Theta}_{m,*}$, and latent parameters $\mathbf{U}\mathbf{V}^\top$ and set horizon $n = 20$, $M_{\text{pre}} = 300000$, $M_{\text{test}} = 200$, $A = 25$, and $d = 5$. Again, the demonstrator $\pi^w$ is the Thomp algorithm. We observe that PreDeToR ($-\tau$) has

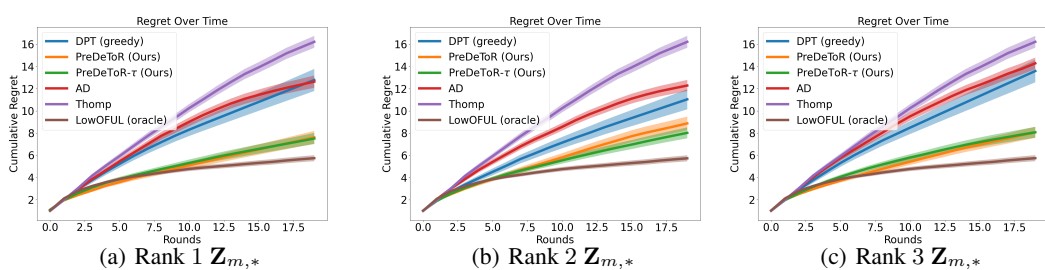

Figure 6: Experiment with latent bandits. The y-axis shows the cumulative regret.

lower cumulative regret than DPT-greedy, AD and Thomp. Note that for any task $m$ for the horizon 20 the Thomp will be able to sample all the actions at most once. Note that for this small horizon setting the DPT-greedy does not have a good estimation of $\widehat{a}_{m,*}$ which results in a poor prediction of optimal action $\widehat{a}_{m,t,*}$. In contrast PreDeToR (-$\tau$) learns the correlation of rewards across tasks and is able to perform well. Observe from Figure 6(a), 6(b), and 6(c) that PreDeToR has lower regret than Thomp and has regret closer to LowOFUL (oracle) which has access to the problem-dependent parameters. Hence. LowOFUL (oracle) outperforms PreDeToR (-$\tau$) in this setting. This shows that PreDeToR is able to exploit the underlying latent structure and reward correlation better than DPT-greedy, and AD.

### A.5 CONNECTION BETWEEN PREDETOR AND LINEAR MULTIVARIATE GAUSSIAN MODEL

In this section, we try to understand the behavior of PreDeToR and its ability to exploit the reward correlation across tasks under a *linear multivariate Gaussian model*. In this model, the hidden task parameter, $\boldsymbol{\theta}_*$, is a random variable drawn from a multi-variate Gaussian distribution (Bishop, 2006) and the feedback follows a linear model. We study this setting since we can estimate the Linear Minimum Mean Square Estimator (LMMSE) in this setting (Carlin & Louis, 2008; Box & Tiao, 2011). This yields a posterior prediction for the mean of each action over all tasks on average, by leveraging the linear structure when $\boldsymbol{\theta}_*$ is drawn from a multi-variate Gaussian distribution. So we can compare the performance of PreDeToR against such an LMMSE and evaluate whether it is exploiting the underlying linear structure and the reward correlation across tasks. We summarize this as follows:

> **Finding 7:** PreDeToR learns the reward correlation covariance matrix from the in-context training data $\mathcal{H}_{\text{train}}$ and acts greedily on it.

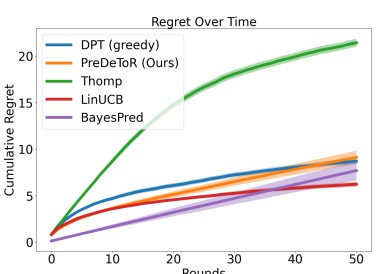

Figure 7: BayesPred Performance

Consider the linear feedback setting consisting of $A$ actions and the hidden task parameter $\boldsymbol{\theta}_* \sim \mathcal{N}(0, \sigma_{\boldsymbol{\theta}}^2 \mathbf{I}_d)$. The reward of the action $\mathbf{x}_t$ at round $t$ is given by $r_t = \mathbf{x}_t^\top \boldsymbol{\theta}_* + \eta_t$, where $\eta_t$ is $\sigma^2$ sub-Gaussian. Let $\pi^w$ collect $n$ rounds of pretraining in-context data and observe $\{I_t, r_t\}_{t=1}^n$. Let $N_n(a)$ denote the total number of times the action $a$ is sampled for $n$ rounds. Note that we drop the task index $m$ in these notations as the random variable $\boldsymbol{\theta}_*$ corresponds to the task. Define the matrix $\mathbf{H}_n \in \mathbb{R}^{n \times A}$ where the $t$-th row represents the action $I_t$ for $t \in [n]$. The $t$-th row of $\mathbf{H}_n$ is a one-hot vector with the $I_t$-th component being 1. We represent each action by one hot vector because we assume that this LMMSE does not have access to the feature vectors of the actions similar to the PreDeToR for fair comparison. Then define the reward vector $\mathbf{Y}_n \in \mathbb{R}^n$ where the $t$-th component is the reward $r_t$ observed for the action $I_t$ for $t \in [n]$ in the pretraining data. Define the diagonal matrix

$\mathbf{D}_A \in \mathbb{R}^{A \times A}$ estimated from pretraining data as follows

$$\mathbf{D}_A(i,i) = \begin{cases} \frac{\sigma^2}{N_n(a)}, & \text{if } N_n(a) > 0 \\ = 0, & \text{if } N_n(a) = 0 \end{cases} \tag{6}$$

where the reward noise being $\sigma^2$ sub-Gaussian is known. Finally define the estimated reward covariance matrix $\mathbf{S}_A \in \mathbb{R}^{A \times A}$ as $\mathbf{S}_A(a, a') = \widehat{\mu}_n(a)\widehat{\mu}_n(a')$, where $\widehat{\mu}_n(a)$ is the empirical mean of action $a$ estimated from the pretraining data. This matrix captures the reward correlation between the pairs of actions $a, a' \in [A]$. Then the posterior average mean estimator $\widehat{\mu} \in \mathbb{R}^A$ over all tasks is given by the following lemma. The proof is given in Appendix B.1.

**Lemma 1.** *Let $\mathbf{H}_n$ be the action matrix, $\mathbf{Y}_n$ be the reward vector and $\mathbf{S}_A$ be the estimated reward covariance matrix. Then the posterior prediction of the average mean reward vector $\widehat{\mu}$ over all tasks is given by*

$$\widehat{\mu} = \sigma_{\boldsymbol{\theta}}^2 \mathbf{S}_A \mathbf{H}_n^\top \left( \sigma_{\boldsymbol{\theta}}^2 \mathbf{H}_n (\mathbf{S}_A + \mathbf{D}_A) \mathbf{H}_n^\top \right)^{-1} \mathbf{Y}_n. \tag{7}$$

The $\widehat{\mu}$ in equation 7 represents the posterior mean vector averaged on all tasks. So if some action $a \in [A]$ consistently yields high rewards in the pretraining data then $\widehat{\mu}(a)$ has high value. Since the test distribution is the same as pretraining, this action on average will yield a high reward during test time.

We hypothesize that the PreDeToR is learning the reward correlation covariance matrix from the training data $\mathcal{H}_{\text{train}}$ and acting greedily on it. To test this hypothesis, we consider the greedy BayesPred algorithm that first estimates $\mathbf{S}_A$ from the pretraining data. It then uses the LMMSE estimator in Lemma 1 to calculate the posterior mean vector $\widehat{\mu}$, and then selects $I_t = \arg\max_a \widehat{\mu}(a)$ at each round $t$. Note that BayesPred is a greedy algorithm that always selects the most rewarding action (exploitation) without any exploration of sub-optimal actions. Also the BayesPred is an LMMSE estimator that leverages the linear reward structure and estimates the reward covariance matrix, and therefore can be interpreted as a lower bound to the regret of PreDeToR. The hypothesis that BayesPred is a lower bound to PreDeToR is supported by Figure 7. In Figure 7 the reward covariance matrix for BayesPred is estimated from the $\mathcal{H}_{\text{train}}$ by first running the Thomp ($\pi^w$). Observe that the BayesPred has a lower cumulative regret than PreDeToR and almost matches the regret of PreDeToR towards the end of the horizon. Also note that LinUCB has lower cumulative regret towards the end of horizon as it leverages the linear structure and the feature of the actions in selecting the next action.

A.6 EMPIRICAL STUDY: INCREASING NUMBER OF ACTIONS

In this section, we discuss the performance of PreDeToR when the number of actions is very high so that the weak demonstrator $\pi^w$ does not have sufficient samples for each action. However, the number of tasks $M_{\text{pre}} \gg A > n$.

**Baselines:** We again implement the same baselines discussed in Section 4. The baselines are PreDeToR, PreDeToR-$\tau$, DPT-greedy, AD, Thomp, and LinUCB.

**Outcomes:** We first discuss the main outcomes from our experimental results of introducing more actions than the horizon (or more dimensions than actions) during data collection and evaluation:

> **Finding 8:** PreDeToR (-$\tau$) outperforms DPT-greedy, and AD, even when $A > n$ but $M_{\text{pre}} \gg A$.

**Experimental Result:** We observe these outcomes in Figure 8. In Figure 8(a) we show the linear bandit setting for $M_{\text{pre}} = 250000$, $M_{\text{test}} = 200$, $A = 100$, $n = 50$ and $d = 5$. Again, the demonstrator $\pi^w$ is the Thomp algorithm. We observe that PreDeToR (-$\tau$) has lower cumulative regret than DPT-greedy and AD. Note that for any task $m$ the Thomp will not be able to sample all the actions even once. The weak performance of DPT-greedy can be attributed to both short horizons and the inability to estimate the optimal action for such a short horizon $n < A$. The AD performs similar to the demonstrator Thomp because of its training. Observe that PreDeToR (-$\tau$) has similar regret to LinUCB and lower regret than Thomp which also shows that PreDeToR is exploiting the

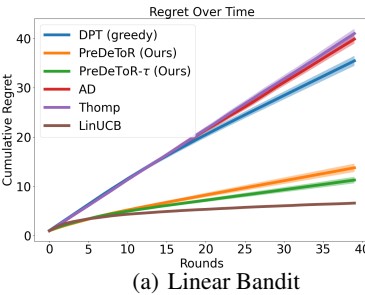 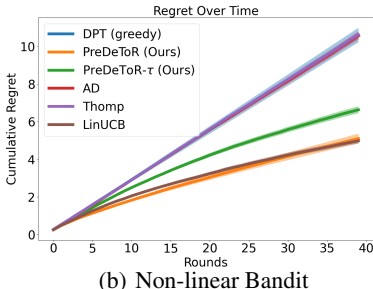

(a) Linear Bandit         (b) Non-linear Bandit

Figure 8: Testing the limit experiments. The horizontal axis is the number of rounds. Confidence bars show one standard error.

latent linear structure of the underlying tasks. In Figure 8(b) we show the non-linear bandit setting for horizon $n = 40$, $M_{\text{pre}} = 200000$, $A = 60$, $d = 2$, and $|\mathcal{A}^{\text{inv}}| = 5$. The demonstrator $\pi^w$ is the Thomp algorithm. Again we observe that PreDeToR (-$\tau$) has lower cumulative regret than DPT-greedy, AD and LinUCB which fails to perform well in this non-linear setting due to its algorithmic design.

A.7 EMPIRICAL STUDY: INCREASING HORIZON

In this section, we discuss the performance of PreDeToR with respect to an increasing horizon for each task $m \in [M]$. However, note that the number of tasks $M_{\text{pre}} \geq n$. Note that Lee et al. (2023) studied linear bandit setting for $n = 200$. We study the setting up to a similar horizon scale.

**Baselines:** We again implement the same baselines discussed in Section 4. The baselines are PreDeToR, PreDeToR-$\tau$, DPT-greedy, AD, Thomp, and LinUCB.

**Outcomes:** We first discuss the main outcomes of our experimental results for increasing the horizon:

> **Finding 9:** PreDeToR (-$\tau$) outperforms DPT-greedy, and AD with increasing horizon.

**Experimental Result:** We observe these outcomes in Figure 9. In Figure 9 we show the linear bandit setting for $M_{\text{pre}} = 150000$, $M_{\text{test}} = 200$, $A = 20$, $n = \{20, 40, 60, 100, 120, 140, 200\}$ and $d = 5$. Again, the demonstrator $\pi^w$ is the Thomp algorithm. We observe that PreDeToR (-$\tau$) has lower cumulative regret than DPT-greedy, and AD. Note that for any task $m$ for the horizon 20 the Thomp will be able to sample all the actions at most once. Observe from Figure 9(a), 9(b), 9(c), Figure 9(d), 9(e), 9(f) and 9(g) that PreDeToR (-$\tau$) is closer to LinUCB and outperforms Thomp which also shows that PreDeToR (-$\tau$) is learning the latent linear structure of the underlying tasks. In Figure 9(h) we plot the regret of all the baselines with respect to the increasing horizon. Again we see that PreDeToR (-$\tau$) is closer to LinUCB and outperforms DPT-greedy, AD and Thomp. This shows that PreDeToR (-$\tau$) is able to exploit the latent structure and reward correlation across the tasks for varying horizon length.

A.8 EMPIRICAL STUDY: INCREASING DIMENSION

In this section, we discuss the performance of PreDeToR with respect to an increasing dimension for each task $m \in [M]$. Again note that the number of tasks $M_{\text{pre}} \gg A \geq n$. Through this experiment, we want to evaluate the performance of PreDeToR and see how it exploits the underlying reward correlation when the horizon is small as well as for increasing dimensions.

**Baselines:** We again implement the same baselines discussed in Section 4. The baselines are PreDeToR, PreDeToR-$\tau$ DPT-greedy, AD, Thomp, and LinUCB.

**Outcomes:** We first discuss the main outcomes of our experimental results for increasing the horizon:

> **Finding 10:** PreDeToR (-$\tau$) outperforms DPT-greedy, AD with increasing dimension and has lower regret than LinUCB for larger dimension.

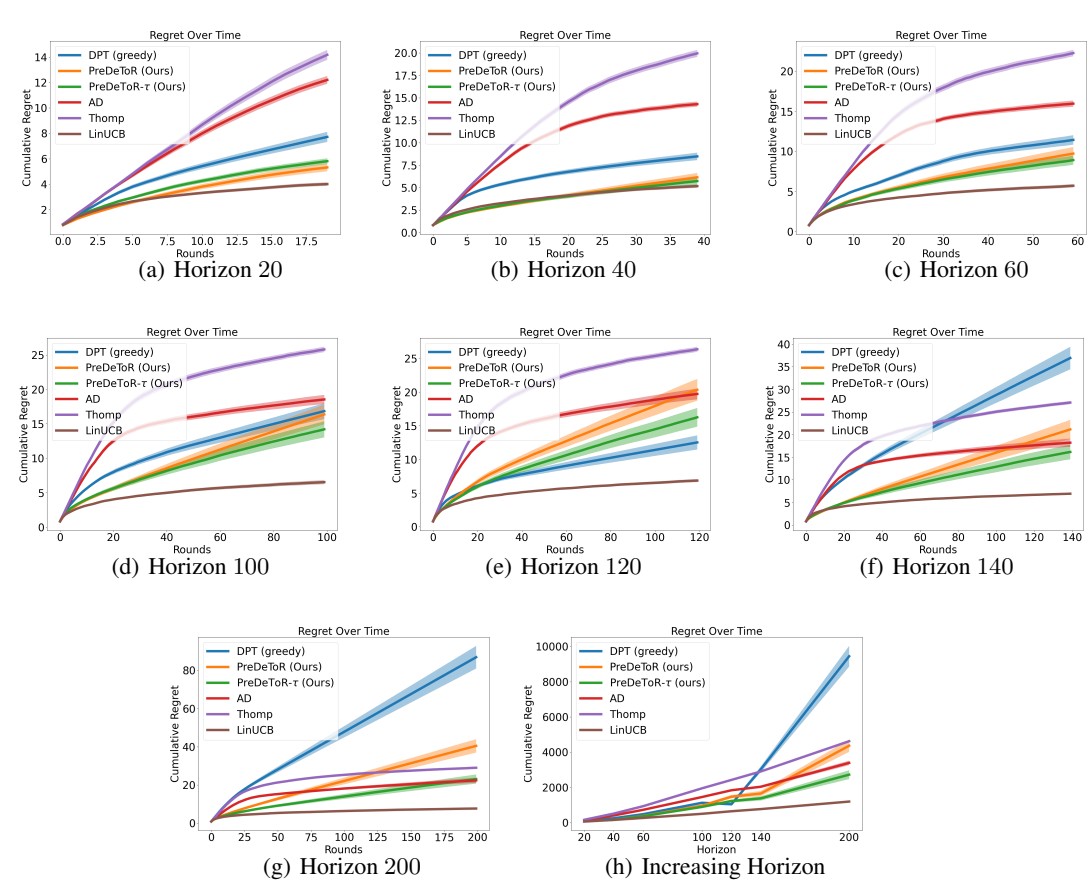

Figure 9: Experiment with increasing horizon. The y-axis shows the cumulative regret.

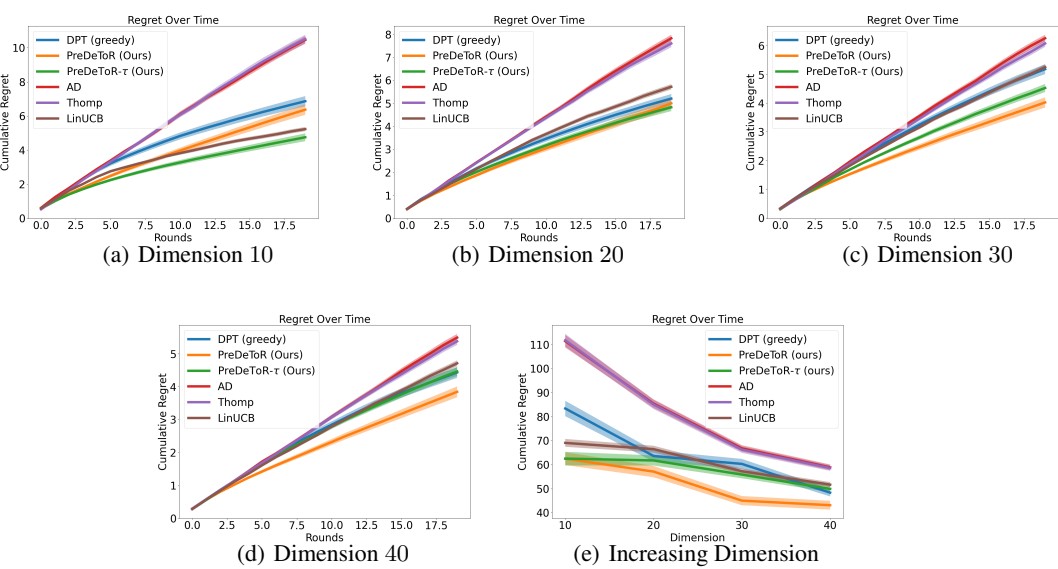

Figure 10: Experiment with increasing dimension. The y-axis shows the cumulative regret.

**Experimental Result:** We observe these outcomes in Figure 9. In Figure 9 we show the linear bandit setting for horizon $n = 20$, $M_{\text{pre}} = 160000$, $M_{\text{test}} = 200$, $A = 20$, and $d = \{10, 20, 30, 40\}$. Again, the demonstrator $\pi^w$ is the Thomp algorithm. We observe that PreDeToR (-$\tau$) has lower cumulative regret than DPT-greedy, AD. Note that for any task $m$ for the horizon 20 the Thomp

will be able to sample all the actions at most once. Observe from Figure 10(a), 10(b), 10(c), and 10(d) that PreDeToR (-$\tau$) is closer to LinUCB and has lower regret than Thomp which also shows that PreDeToR (-$\tau$) is exploiting the latent linear structure of the underlying tasks. In Figure 10(e) we plot the regret of all the baselines with respect to the increasing dimension. Again we see that PreDeToR (-$\tau$) has lower regret than DPT-greedy, AD and Thomp. Observe that with increasing dimension PreDeToR is able to outperform LinUCB. This shows that the PreDeToR (-$\tau$) is able to exploit reward correlation across tasks for varying dimensions.

### A.9 EMPIRICAL STUDY: INCREASING ATTENTION HEADS

In this section, we discuss the performance of PreDeToR with respect to an increasing attention heads for the transformer model for the non-linear feedback model. Again note that the number of tasks $M_{\text{pre}} \gg A \geq n$. Through this experiment, we want to evaluate the performance of PreDeToR to exploit the underlying reward correlation when the horizon is small and understand the representative power of the transformer by increasing the attention heads. Note that we choose the non-linear feedback model and low data regime to leverage the representative power of the transformer.

**Baselines:** We again implement the same baselines discussed in Section 4. The baselines are PreDeToR, PreDeToR-$\tau$, DPT-greedy, AD, Thomp, and LinUCB.

**Outcomes:** We first discuss the main outcomes of our experimental results for increasing the horizon:

> **Finding 11:** PreDeToR (-$\tau$) outperforms DPT-greedy, and AD with increasing attention heads.

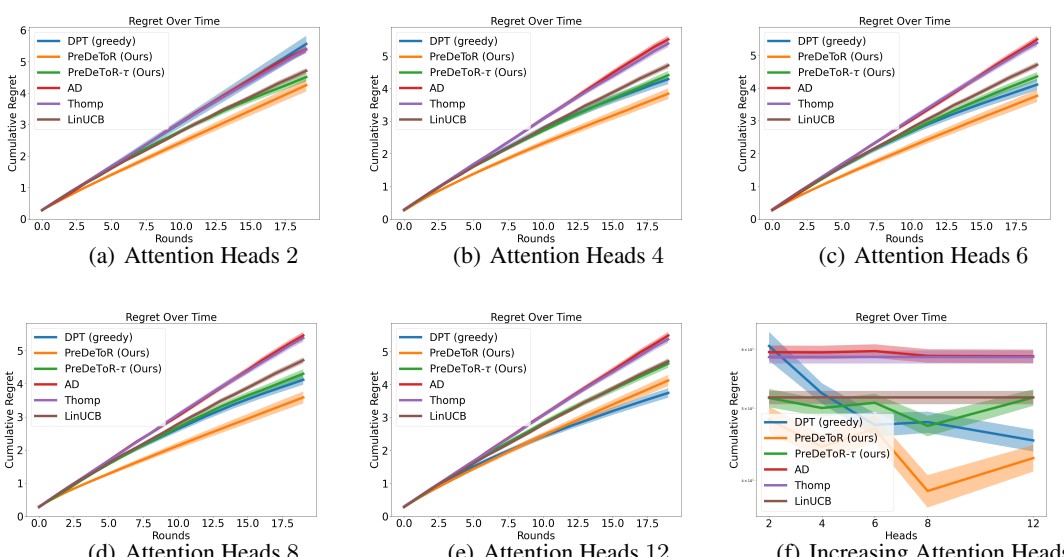

Figure 11: Experiment with increasing attention heads. The y-axis shows the cumulative regret.

**Experimental Result:** We observe these outcomes in Figure 11. In Figure 11 we show the non-linear bandit setting for horizon $n = 20$, $M_{\text{pre}} = 160000$, $M_{\text{test}} = 200$, $A = 20$, heads $= \{2, 4, 6, 8\}$ and $d = 5$. Again, the demonstrator $\pi^w$ is the Thomp algorithm. We observe that PreDeToR (-$\tau$) has lower cumulative regret than DPT-greedy, AD. Note that for any task $m$ for the horizon 20 the Thomp will be able to sample all the actions atmost once. Observe from Figure 11(a), 11(b), 11(c), and 11(d) that PreDeToR (-$\tau$) has lower regret than AD, Thomp and LinUCB which also shows that PreDeToR (-$\tau$) is exploiting the latent linear structure of the underlying tasks for the non-linear setting. In Figure 11(f) we plot the regret of all the baselines with respect to the increasing attention heads. Again we see that PreDeToR (-$\tau$) regret decreases as we increase the attention heads.

## A.10 EMPIRICAL STUDY: INCREASING NUMBER OF TASKS

In this section, we discuss the performance of PreDeToR with respect to the increasing number of tasks for the linear bandit setting. Again note that the number of tasks $M_{\text{pre}} \gg A \geq n$. Through this experiment, we want to evaluate the performance of PreDeToR to exploit the underlying reward correlation when the horizon is small and the number of tasks is changing. Finally, recall that when the horizon is small the weak demonstrator $\pi^w$ does not have sufficient samples for each action. This leads to a poor approximation of the greedy action.

**Baselines:** We again implement the same baselines discussed in Section 4. The baselines are PreDeToR, PreDeToR-$\tau$, DPT-greedy, AD, Thomp, and LinUCB.

**Outcomes:** We first discuss the main outcomes of our experimental results for increasing the horizon:

> **Finding 12:** PreDeToR (-$\tau$) fails to exploit the underlying latent structure and reward correlation from in-context data when the number of tasks is small.

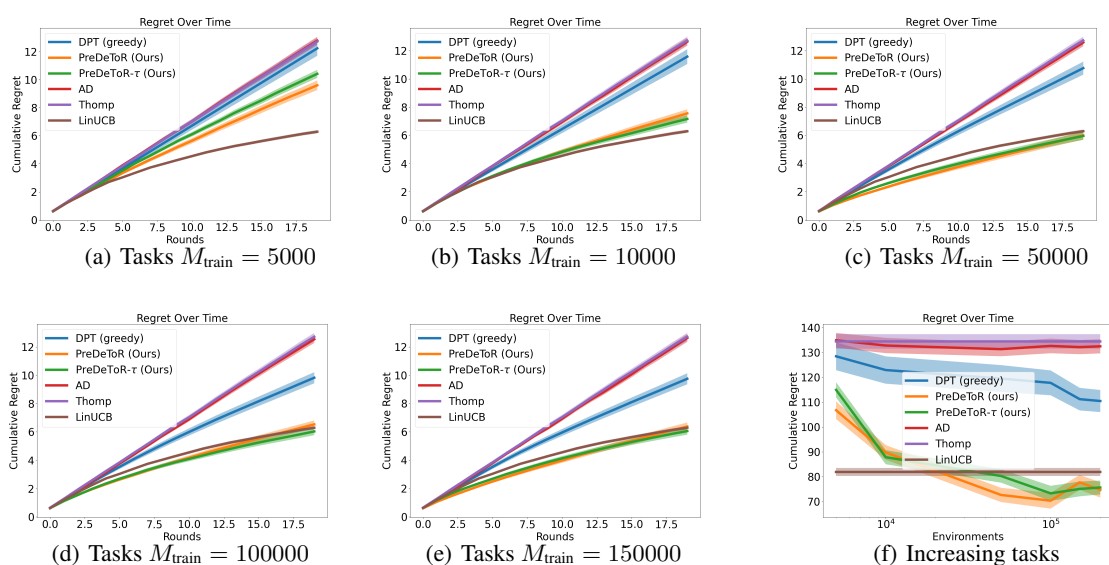

(a) Tasks $M_{\text{train}} = 5000$  (b) Tasks $M_{\text{train}} = 10000$  (c) Tasks $M_{\text{train}} = 50000$

(d) Tasks $M_{\text{train}} = 100000$  (e) Tasks $M_{\text{train}} = 150000$  (f) Increasing tasks

Figure 12: Experiment with an increasing number of tasks. The y-axis shows the cumulative regret.

**Experimental Result:** We observe these outcomes in Figure 12. In Figure 12 we show the linear bandit setting for horizon $n = 20$, $M_{\text{pre}} \in \{5000, 10000, 50000, 100000, 150000\}$, $M_{\text{test}} = 200$, $A = 20$, and $d = 40$. Again, the demonstrator $\pi^w$ is the Thomp algorithm. We observe that PreDeToR (-$\tau$), AD and DPT-greedy suffer more regret than the LinUCB when the number of tasks is small ($M_{\text{train}} \in \{5000, 10000\}$ in Figure 12(a), and 12(b). However in Figure 12(c), 12(d), 12(e), and 12(f) we show that PreDeToR has lower regret than Thomp and matches LinUCB. This shows that PreDeToR (-$\tau$) is exploiting the latent linear structure of the underlying tasks for the non-linear setting. Moreover, observe that as $M_{\text{train}}$ increases the PreDeToR has lower cumulative regret than DPT-greedy, AD. Note that for any task $m$ for the horizon 20 the Thomp will be able to sample all the actions at most once. Therefore DPT-greedy does not perform as well as PreDeToR. Finally, note that the result shows that PreDeToR (-$\tau$) is able to exploit the reward correlation across the tasks better as the number of tasks increases.

## A.11 EXPLORATION OF PREDETOR(-$\tau$)

In this section, we discuss the exploration of PreDeToR in the linear bandit setting discussed in Section 4. Recall that the linear bandit setting consist of horizon $n = 25$, $M_{\text{pre}} = 200000$, $M_{\text{test}} = 200$, $A = 10$, and $d = 2$. The demonstrator $\pi^w$ is the Thomp algorithm and we observe that PreDeToR (-$\tau$) has lower cumulative regret than DPT-greedy, AD and matches the performance of

LinUCB. Therefore PreDeToR (-$\tau$) behaves almost optimally in this setting and so we analyze how PreDeToR conducts exploration for this setting.

**Outcomes:** We first discuss the main outcomes of our analysis of exploration in the low-data regime:

> **Finding 13:** The PreDeToR (-$\tau$) has a two phase exploration. In the first phase, it explores with a strong prior over the in-context training data. In the second phase, once the task data has been observed for a few rounds (in-context) it switches to task-based exploration.

We first show in Figure 13(a) the training distribution of the optimal actions. For each bar, the frequency indicates the number of tasks where the action (shown in the x-axis) is the optimal action.

Then in Figure 13(b) we show how the sampling distribution of DPT-greedy, PreDeToR and PreDeToR-$\tau$ change in the first 10 and last 10 rounds for all the tasks where action 5 is optimal. To plot this graph we first sum over the individual pulls of the action taken by each algorithm over the first 10 and last 10 rounds. Then we average these counts over all test tasks where action 5 is optimal. From the figure Figure 13(b) we see that PreDeToR(-$\tau$) consistently pulls the action 5 more than DPT-greedy. It also explores other optimal actions like $\{2, 3, 6, 7, 10\}$ but discards them quickly in favor of the optimal action 5 in these tasks. This shows that PreDeToR (-$\tau$) only considers the optimal actions seen from the training data. Once sufficient observation have been observed for the task it switches to task-based exploration and samples the optimal action more than DPT-greedy.

Finally, we plot the feasible action set considered by DPT-greedy, PreDeToR, and PreDeToR-$\tau$ in Figure 13(c). To plot this graph again we consider the test tasks where the optimal action is 5. Then we count the number of distinct actions that are taken from round $t$ up until horizon $n$. Finally we average this over all the considered tasks where the optimal action is 5. We call this the candidate action set considered by the algorithm. From the Figure 13(c) we see that DPT-greedy explores the least and gets stuck with few actions quickly (by round 10). Note that the actions DPT-greedy samples are sub-optimal and so it suffers a high cumulative regret (see Figure 2). PreDeToR explore slightly more than DPT-greedy, but PreDeToR-$\tau$ explores the most.

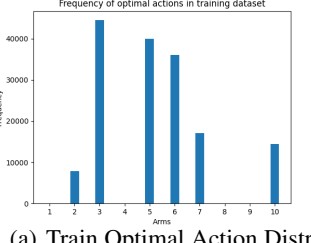
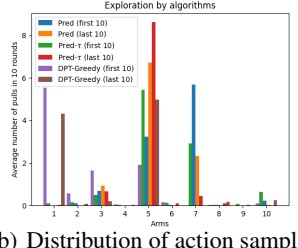
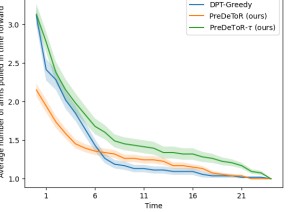

(a) Train Optimal Action Distribution

(b) Distribution of action sampling in all tasks where action 5 is optimal

(c) Candidate Action Set in Time averaged over all tasks where action 5 is optimal

Figure 13: Exploration Analysis of PreDeToR(-$\tau$)

## A.12 EXPLORATION OF PREDETOR(-$\tau$) IN NEW ARMS SETTING

In this section, we discuss the exploration of PreDeToR (-$\tau$) in the linear and non-linear new arms bandit setting discussed in Section 6. Recall that we consider the linear bandit setting of horizon $n = 50$, $M_{\text{pre}} = 200000$, $M_{\text{test}} = 200$, $A = 20$, and $d = 5$. Here during data collection and during collecting the test data, we randomly select one new action from $\mathbb{R}^d$ for each task $m$. So the number of invariant actions is $|\mathcal{A}^{\text{inv}}| = 19$.

**Outcomes:** We first discuss the main outcomes of our analysis of exploration in the low-data regime:

**Finding 14:** The PreDeToR (-$\tau$) is robust to changes when the number of in-variant actions is large. PreDeToR (-$\tau$) performance drops as shared structure breaks down.

We first show in Figure 14(a) the training distribution of the optimal actions. For each bar, the frequency indicates the number of tasks where the action (shown in the x-axis) is the optimal action.

Then in Figure 14(b) we show how the sampling distribution of DPT-greedy, PreDeToR and PreDeToR-$\tau$ change in the first 10 and last 10 rounds for all the tasks where action 17 is optimal. We plot this graph the same way as discussed in Appendix A.11. From the figure Figure 14(b) we see that PreDeToR(-$\tau$) consistently pulls the action 17 more than DPT-greedy. It also explores other optimal actions like $\{1, 2, 3, 8, 9, 15\}$ but discards them quickly in favor of the optimal action 17 in these tasks.

Finally, we plot the feasible action set considered by DPT-greedy, PreDeToR, and PreDeToR-$\tau$ in Figure 14(c). To plot this graph again we consider the test tasks where the optimal action is 17. Then we count the number of distinct actions that are taken from round $t$ up until horizon $n$. Finally we average this over all the considered tasks where the optimal action is 17. We call this the candidate action set considered by the algorithm. From the Figure 14(c) we see that PreDeToR-$\tau$ explores more than PreDeToR in this setting.

We also show how the prediction error of the optimal action by PreDeToR compared to LinUCB in this 1 new arm linear bandit setting. In Figure 15(a) we first show how the 20 actions are distributed in the $M_{\text{test}} = 200$ test tasks. In Figure 15(a) for each bar, the frequency indicates the number of tasks where the action (shown in the x-axis) is the optimal action. Then in Figure 15(b) we show the prediction error of PreDeToR (-$\tau$) for each task $m \in [M_{\text{test}}]$. The prediction error is calculated the same way as stated in Section 6 From the Figure 15(b) we see that for most actions the prediction error of PreDeToR (-$\tau$) is closer to LinUCB showing that the introduction of 1 new action does not alter the prediction error much. Note that LinUCB estimates the empirical mean directly from the test task, whereas PreDeToR has a strong prior based on the training data. Therefore we see that PreDeToR is able to estimate the reward of the optimal action quite well from the training dataset $\mathcal{D}_{\text{pre}}$.

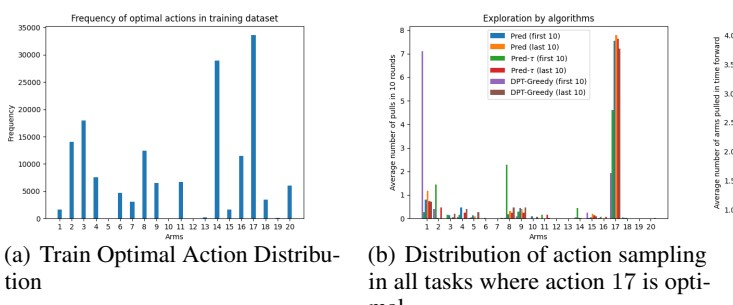 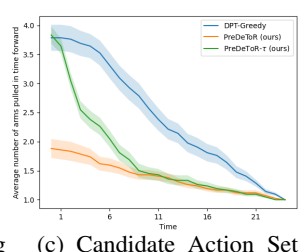

(a) Train Optimal Action Distribution

(b) Distribution of action sampling in all tasks where action 17 is optimal

(c) Candidate Action Set in Time averaged over all tasks where action 17 is optimal

Figure 14: Exploration Analysis of PreDeToR(-$\tau$) in linear 1 new arm setting

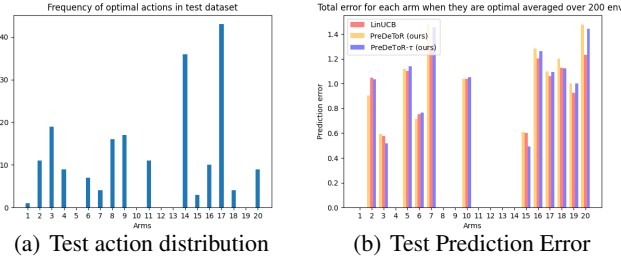

(a) Test action distribution

(b) Test Prediction Error

Figure 15: Prediction error of PreDeToR(-$\tau$) in linear 1 new arm setting

We now consider the setting where the number of invariant actions is $|\mathcal{A}^{\text{inv}}| = 15$. We again show in Figure 16(a) the training distribution of the optimal actions. For each bar, the frequency indicates the number of tasks where the action (shown in the x-axis) is the optimal action. Then in Figure 16(b) we show how the sampling distribution of DPT-greedy, PreDeToR and PreDeToR-$\tau$ change in the first 10 and last 10 rounds for all the tasks where action 17 is optimal. We plot this graph the same way as discussed in Appendix A.11. From the figure Figure 16(b) we see that none of the algorithms PreDeToR, PreDeToR-$\tau$, DPT-greedy consistently pulls the action 17 more than other actions. This shows that the common underlying actions across the tasks matter for learning the epxloration.

Finally, we plot the feasible action set considered by DPT-greedy, PreDeToR, and PreDeToR-$\tau$ in Figure 16(c). To plot this graph again we consider the test tasks where the optimal action is 17. We build the candidate set the same way as before. From the Figure 16(c) we see that none of the three algorithms DPT-greedy, PreDeToR, PreDeToR-$\tau$, is able to sample the optimal action 17 sufficiently high number of times.

We also show how the prediction error of the optimal action by PreDeToR compared to LinUCB in this 1 new arm linear bandit setting. In Figure 17(a) we first show how the 20 actions are distributed in the $M_{\text{test}} = 200$ test tasks. In Figure 17(a) for each bar, the frequency indicates the number of tasks where the action (shown in the x-axis) is the optimal action. Then in Figure 17(b) we show the prediction error of PreDeToR (-$\tau$) for each task $m \in [M_{\text{test}}]$. The prediction error is calculated the same way as stated in Section 6. From the Figure 17(b) we see that for most actions the prediction error is higher than LinUCB showing that the introduction of 5 new actions (and thereby decreasing the invariant action set) significantly alters the prediction error.

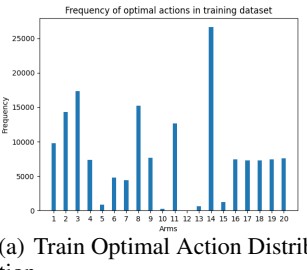 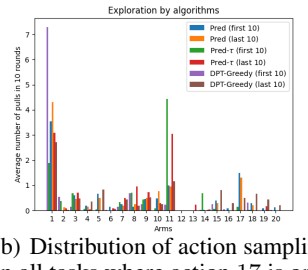 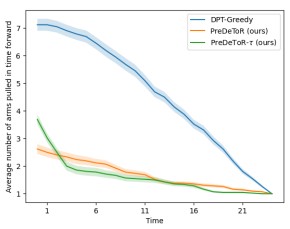

(a) Train Optimal Action Distribution

(b) Distribution of action sampling in all tasks where action 17 is optimal

(c) Candidate Action Set in Time averaged all tasks where action 17 is optimal

Figure 16: Exploration Analysis of PreDeToR(-$\tau$) in linear 5 new arm setting

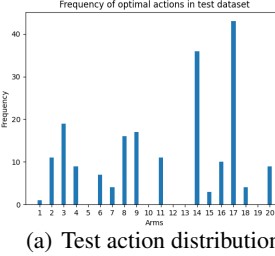 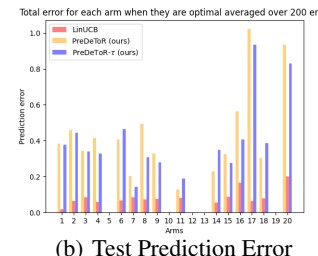

(a) Test action distribution

(b) Test Prediction Error

Figure 17: Prediction error of PreDeToR(-$\tau$) in linear 1 new arm setting

## A.13 DATA COLLECTION ANALYSIS

In this section, we analyze the performance of PreDeToR, PreDeToR-$\tau$, DPT-greedy, AD, Thomp, and LinUCB when the weak demonstrator $\pi^w$ is Thomp, LinUCB, or Uniform. We again consider the linear bandit setting discussed in Section 4. Recall that the linear bandit setting consist of horizon $n = 25$, $M_{\text{pre}} = 200000$, $M_{\text{test}} = 200$, $A = 10$, and $d = 2$. Finally, we show the cumulative regret by the above baselines in Figure 18(a), 18(b), and 18(b) when data is collected through Thomp, LinUCB, and Uniform respectively.

**Outcomes:** We first discuss the main outcomes of our experimental results for different data collection:

> **Finding 15:** The PreDeToR (-$\tau$) excels in exploiting the underlying latent structure and reward correlation from in-context data when the data diversity is high.

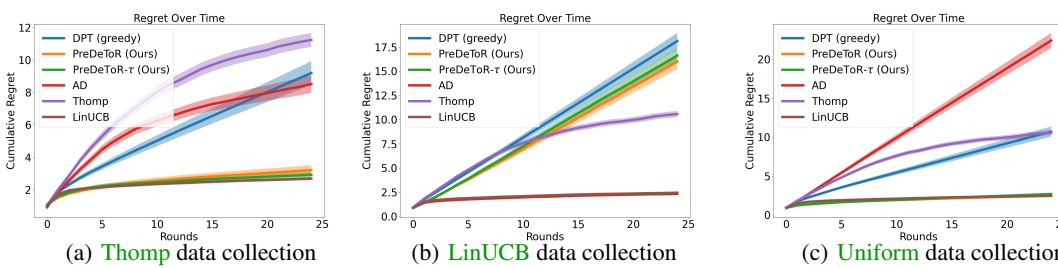

(a) Thomp data collection      (b) LinUCB data collection      (c) Uniform data collection

Figure 18: Data Collection with various algorithms and Performance analysis

**Experimental Result:** We observe these outcomes in Figure 18. In Figure 18(a) we see that the $A$-actioned Thomp is explorative enough as it does not explore with the knowledge of feature representation. So it pulls the sub-optimal actions sufficiently high number of times before discarding them in favor of the optimal action. Therefore the training data is diverse enough so that PreDeToR (-$\tau$) can predict the reward vectors for actions sufficiently well. Consequently, PreDeToR (-$\tau$) almost matches the LinUCB algorithm. Both DPT-greedy and ADperform poorly in this setting.

In Figure 18(b) we see that the LinUCB algorithm is not explorative enough as it explores with the knowledge of feature representation and quickly discards the sub-optimal actions in favor of the optimal action. Therefore the training data is not diverse enough so that PreDeToR (-$\tau$) is not able to correctly predict the reward vectors for actions. Note that DPT-greedy also performs poorly in this setting when it is not provided with the optimal action information during training. The AD matches the performance of its demonstrator LinUCB because of its training procedure of predicting the next action of the demonstrator.

Finally, in Figure 18(c) we see that the $A$-armed Uniform is fully explorative as it does not intend to minimize regret (as opposed to Thomp) and does not explore with the knowledge of feature representation. Therefore the training data is very diverse which results in PreDeToR (-$\tau$) being able to predict the reward vectors for actions very well. Consequently, PreDeToR (-$\tau$) perfectly matches the LinUCB algorithm. Note that AD performs the worst as it matches the performance of its demonstrator whereas the performance of DPT-greedy suffers due to the lack of information on the optimal action during training.

### A.14 EMPIRICAL VALIDATION OF THEORETICAL RESULT

In this section, we empirically validate the theoretical result proved in Section 7. We again consider the linear bandit setting discussed in Section 4. Recall that the linear bandit setting consist of horizon $n = 25$, $M_{pre} = \{100000, 200000\}$, $M_{test} = 200$, $A = 10$, and $d = 2$. The demonstrator $\pi^w$ is the Thomp algorithm and we observe that PreDeToR (-$\tau$) has lower cumulative regret than DPT-greedy, AD and matches the performance of LinUCB.

**Baseline (LinUCB-$\tau$):** We define soft LinUCB (LinUCB-$\tau$) as follows: At every round $t$ for task $m$, it calculates the ucb value $B_{m,a,t}$ for each action $\mathbf{x}_{m,a} \in \mathcal{X}$ such that $B_{m,a,t} = \mathbf{x}_{m,a}^\top \widehat{\boldsymbol{\theta}}_{m,t-1} + \alpha \|\mathbf{x}_{m,a}\|_{\boldsymbol{\Sigma}_{m,t-1}^{-1}}$ where $\alpha > 0$ is a constant and $\widehat{\boldsymbol{\theta}}_{m,t}$ is the estimate of the model parameter $\boldsymbol{\theta}_{m,*}$ at round $t$. Here, $\boldsymbol{\Sigma}_{m,t-1} = \sum_{s=1}^{t-1} \mathbf{x}_{m,s} \mathbf{x}_{m,s}^\top + \lambda \mathbf{I}_d$ is the data covariance matrix or the arms already tried. Then it chooses $I_t \sim \mathrm{softmax}_a^\tau(B_{m,a,t})$, where $\mathrm{softmax}_a^\tau(\cdot) \in \triangle^A$ denotes a softmax

distribution over the actions and $\tau$ is a temperature parameter (See Section 4 for definition of $\mathrm{softmax}_a^\tau(\cdot)$).

**Outcomes:** We first discuss the main outcomes of our experimental results:

---

**Finding 16:** PreDeToR ($-\tau$) excels in predicting the rewards for test tasks when the number of training (source) tasks is large.

---

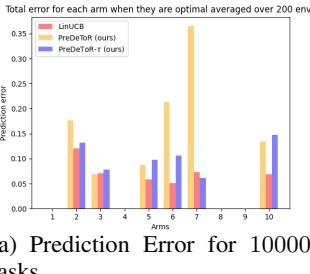 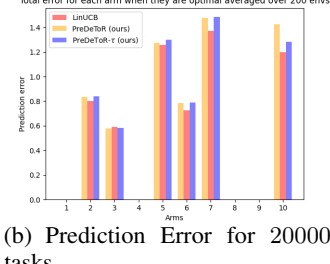 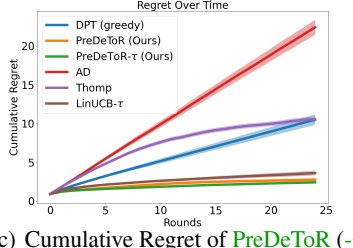

(a) Prediction Error for 100000 tasks     (b) Prediction Error for 200000 tasks     (c) Cumulative Regret of PreDeToR ($-\tau$) compared against LinUCB-$\tau$

Figure 19: Empirical validation of theoretical analysis

**Experimental Result:** These findings are reported in Figure 19. In Figure 19(a) we show the prediction error of PreDeToR ($-\tau$) for each task $m \in [M_\text{test}]$. The prediction error is calculated as $(\widehat{\mu}_{m,n,*}(a) - \mu_{m,*}(a))^2$ where $\widehat{\mu}_{m,n,*}(a) = \max_a \widehat{\boldsymbol{\theta}}_{m,n}^\top \mathbf{x}_m(a)$ is the empirical mean at the end of round $n$, and $\mu_{*,m}(a) = \max_a \boldsymbol{\theta}_{m,*}^\top \mathbf{x}_m(a)$ is the true mean of the optimal action in task $m$. Then we average the prediction error for the action $a \in [A]$ by the number of times the action $a$ is the optimal action in some task $m$. We see that when the source tasks are 100000 the reward prediction falls short of LinUCB prediction for all actions except action 2.

In Figure 19(b) we again show the prediction error of PreDeToR ($-\tau$) for each task $m \in [M_\text{test}]$ when the source tasks are 200000. Note that in both these settings, we kept the horizon $n = 25$, and the same set of actions. We now observe that the reward prediction almost matches LinUCB prediction in almost all the optimal actions.

In Figure 19(c) we compare PreDeToR ($-\tau$) against LinUCB-$\tau$ and show that they almost match in the linear bandit setting discussed in Section 4 when the source tasks are 100000.

A.15   EMPIRICAL STUDY: OFFLINE PERFORMANCE

In this section, we discuss the offline performance of PreDeToR when the number of tasks $M_\text{pre} \gg A \geq n$.

We first discuss how PreDeToR ($-\tau$) is modified for the offline setting. In the offline setting, the PreDeToR first samples a task $m \sim \mathcal{T}_\text{test}$, then the test dataset $\mathcal{H}_m \sim \mathcal{D}_\text{test}(\cdot|m)$. Then PreDeToR and PreDeToR-$\tau$ act similarly to the online setting, but based on the entire offline dataset $\mathcal{H}_m$. The full pseudocode of PreDeToR is in Algorithm 2.

Recall that $\mathcal{D}_\text{test}$ denote a distribution over all possible interactions that can be generated by $\pi^w$ during test time. For offline testing, first, a test task $m \sim \mathcal{T}_\text{test}$, and then an in-context test dataset $\mathcal{H}_m$ is collected such that $\mathcal{H}_m \sim \mathcal{D}_\text{test}(\cdot|m)$. Observe from Algorithm 2 that in the offline setting, PreDeToR first samples a task $m \sim \mathcal{T}_\text{test}$, and then a test dataset $\mathcal{H}_m \sim \mathcal{D}_\text{test}(\cdot|m)$ and acts greedily. Crucially in the offline setting the PreDeToR does not add the observed reward $r_t$ at round $t$ to the dataset. Through this experiment, we want to evaluate the performance of PreDeToR to learn the underlying latent structure and reward correlation when the horizon is small. Finally, recall that when the horizon is small the weak demonstrator $\pi^w$ does not have sufficient samples for each action. This leads to a poor approximation of the greedy action.

---

**Algorithm 2 Pre-trained Decision Transformer with Reward Estimation (PreDeToR)**

1: **Collecting Pretraining Dataset**
2: Initialize empty pretraining dataset $\mathcal{H}_{\text{train}}$
3: **for** $i$ in $[M_{\text{pre}}]$ **do**
4:     Sample task $m \sim \mathcal{T}_{\text{pre}}$, in-context dataset $\mathcal{H}_m \sim \mathcal{D}_{\text{pre}}(\cdot|m)$ and add this to $\mathcal{H}_{\text{train}}$.
5: **end for**
6: **Pretraining model on dataset**
7: Initialize model $\text{TF}_{\Theta}$ with parameters $\Theta$
8: **while** not converged **do**
9:     Sample $\mathcal{H}_m$ from $\mathcal{H}_{\text{train}}$ and predict $\widehat{r}_{m,t}$ for action $(I_{m,t})$ for all $t \in [n]$
10:     Compute loss in equation 3 with respect to $r_{m,t}$ and backpropagate to update model parameter $\Theta$.
11: **end while**
12: **Offline test-time deployment**
13: Sample unknown task $m \sim \mathcal{T}_{\text{test}}$, sample dataset $\mathcal{H}_m \sim \mathcal{D}_{\text{test}}(\cdot|m)$
14: Use $\text{TF}_{\Theta}$ on $m$ at round $t$ to choose

$$I_t \begin{cases} = \arg\max_{a \in \mathcal{A}} \text{TF}_{\Theta}\left(\widehat{r}_{m,t}(a) \mid \mathcal{H}_m\right), & \textbf{PreDeToR} \\ \sim \text{softmax}_a^{\tau} \text{TF}_{\Theta}\left(\widehat{r}_{m,t}(a) \mid \mathcal{H}_m\right), & \textbf{PreDeToR-}\tau \end{cases}$$

---

**Baselines:** We again implement the same baselines discussed in Section 4. The baselines are PreDeToR, PreDeToR-$\tau$, DPT-greedy, AD, Thomp, and LinUCB. During test time evaluation for offline setting the DPT selects $I_t = \widehat{a}_{m,t,*}$ where $\widehat{a}_{m,t,*} = \arg\max_a \text{TF}_{\Theta}(a|\mathcal{H}_m^t)$ is the predicted optimal action.

**Outcomes:** We first discuss the main outcomes of our experimental results for increasing the horizon:

> **Finding 17:** PreDeToR (-$\tau$) performs comparably to DPT-greedy and AD in the offline setting.

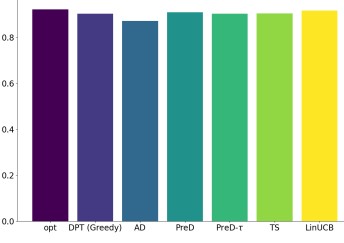
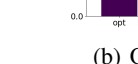

      (a) Offline for Linear setting        (b) Offline for Non-linear setting

Figure 20: Offline experiment. The y-axis shows the cumulative reward.

**Experimental Result:** We observe these outcomes in Figure 20. In Figure 20 we show the linear bandit setting for horizon $n = 20$, $M_{\text{pre}} = 200000$, $M_{\text{test}} = 5000$, $A = 20$, and $d = 5$ for the low data regime. Again, the demonstrator $\pi^w$ is the Thomp algorithm. We observe that PreDeToR (-$\tau$) has comparable cumulative regret to DPT-greedy. Note that for any task $m$ for the horizon $n = 20$ the Thomp will be able to sample all the actions at most once. In the non-linear setting of Figure 20(b) the $n = 40$, $M_{\text{pre}} = 100000$, $A = 6$, $d = 2$. Observe that in all of these results, the performance of PreDeToR (-$\tau$) is comparable with respect to cumulative regret against DPT-greedy.

## B    THEORETICAL ANALYSIS

### B.1    PROOF OF LEMMA 1

*Proof.* The learner collects $n$ rounds of data following $\pi^w$. The weak demonstrator $\pi^w$ only observes the $\{I_t, r_t\}_{t=1}^n$. Recall that $N_n(a)$ denotes the total number of times the action $a$ is sampled for $n$ rounds. Define the matrix $\mathbf{H}_n \in \mathbb{R}^{n \times A}$ where the $t$-th row represents the action sampled at round $t \in [n]$. The $t$-th row is a one-hot vector with 1 as the $a$-th component in the vector for $a \in [A]$. Then define the reward vector $\mathbf{Y}_n \in \mathbb{R}^n$ as the reward vector where the $t$-th component is the observed reward for the action $I_t$ for $t \in [n]$. Finally define the diagonal matrix $\mathbf{D}_A \in \mathbb{R}^{A \times A}$ as in equation 6 and the estimated reward covariance matrix as $\mathbf{S}_A \in \mathbb{R}^{A \times A}$ such that $\mathbf{S}_A(a, a') = \widehat{\mu}_n(a)\widehat{\mu}_n(a')$. This matrix captures the reward correlation between the pairs of actions $a, a' \in [A]$.

Assume $\mu \sim \mathcal{N}(0, \mathbf{S}_*)$ where $\mathbf{S}_* \in \mathbb{R}^{A \times A}$. Then the observed mean vector $\mathbf{Y}_n$ is

$$\mathbf{Y}_n = \mathbf{H}_n \mu + \mathbf{H}_n \mathbf{D}_A^{1/2} \eta_n$$

where, $\eta_n$ is the noise vector over the $[n]$ training data. Then the posterior mean of $\widehat{\mu}$ by Gauss Markov Theorem (Johnson et al., 2002) is given by

$$\widehat{\mu} = \mathbf{S}_* \mathbf{H}_n^\top \left( \mathbf{H}_n (\mathbf{S}_* + \mathbf{D}_A) \mathbf{H}_n^\top \right)^{-1} \mathbf{Y}_n. \tag{8}$$

However, the learner does not know the true reward co-variance matrix. Hence it needs to estimate the $\mathbf{S}_*$ from the observed data. Let the estimate be denoted by $\mathbf{S}_A$.

**Assumption B.1.** We assume that $\pi^w$ is sufficiently exploratory so that each action is sampled at least once.

The Assumption B.1 ensures that the matrix $\left( \sigma_{\boldsymbol{\theta}}^2 \mathbf{H}_n (\mathbf{S}_A + \mathbf{D}_A) \mathbf{H}_n^\top \right)^{-1}$ is invertible. Under Assumption B.1, plugging the estimate $\mathbf{S}_A$ back in equation 8 shows that the average posterior mean over all the tasks is

$$\widehat{\mu} = \mathbf{S}_A \mathbf{H}_n^\top \left( \mathbf{H}_n (\mathbf{S}_A + \mathbf{D}_A) \mathbf{H}_n^\top \right)^{-1} \mathbf{Y}_n. \tag{9}$$

The claim of the lemma follows. □

## C    GENERALIZATION AND TRANSFER LEARNING PROOF FOR PREDETOR

### C.1    GENERALIZATION PROOF

Alg is the space of algorithms induced by the transformer $\mathrm{TF}_\Theta$.

**Theorem C.1.** *(PreDeToR risk) Suppose error stability Assumption 7.1 holds and assume loss function $\ell(\cdot, \cdot)$ is $C$-Lipschitz for all $r_t \in [0, B]$ and horizon $n \geq 1$. Let $\widehat{\mathrm{TF}}$ be the empirical solution of (ERM) and $\mathcal{N}(\mathcal{A}, \rho, \epsilon)$ be the covering number of the algorithm space* Alg *following Definition C.2 and C.3. Then with probability at least $1 - 2\delta$, the excess Multi-task learning (MTL) risk of PreDeToR-$\tau$ is bounded by*

$$\mathcal{R}_{\mathrm{MTL}}(\widehat{\mathrm{TF}}) \leq 4 \frac{C}{\sqrt{nM}} + 2(B + K \log n) \sqrt{\frac{\log(\mathcal{N}(\mathrm{Alg}, \rho, \varepsilon)/\delta)}{cnM}}$$

*where, $\mathcal{N}(\mathrm{Alg}, \rho, \varepsilon)$ is the covering number of transformer $\widehat{\mathrm{TF}}$.*

*Proof.* We consider a meta-learning setting. Let $M$ source tasks are i.i.d. sampled from a task distribution $\mathcal{T}$, and let $\widehat{\mathrm{TF}}$ be the empirical Multitask (MTL) solution. Define $\mathcal{H}_{\mathrm{all}} = \bigcup_{m=1}^M \mathcal{H}_m$. We drop the $\Theta, \mathbf{r}$ from transformer notation $\mathrm{TF}^{\mathbf{r}}_\Theta$ as we keep the architecture fixed as in Lin et al. (2023). Note that this transformer predicts a reward vector over the actions. To be more precise we denote the reward predicted by the transformer at round $t$ after observing history $\mathcal{H}_m^{t-1}$ and then sampling the action $a_{mt}$ as $\mathrm{TF}\left( \widehat{r}_{mt}(a_{mt}) | \mathcal{H}_m^{t-1}, a_{mt} \right)$. Define the training risk

$$\widehat{\mathcal{L}}_{\mathcal{H}_{\mathrm{all}}}(\mathrm{TF}) = \frac{1}{nM} \sum_{m=1}^M \sum_{t=1}^n \ell\left( r_{mt}(a_{mt}), \mathrm{TF}\left( \widehat{r}_{mt}(a_{mt}) | \mathcal{H}_m^{t-1}, a_{mt} \right) \right)$$

and the test risk

$$\mathcal{L}_{\mathrm{MTL}}(\mathrm{TF}) = \mathbb{E}\left[\widehat{\mathcal{L}}_{\mathcal{H}_{\mathrm{all}}}(\mathrm{TF})\right].$$

Define empirical risk minima $\widehat{\mathrm{TF}} = \arg\min_{\mathrm{TF}\in\mathrm{Alg}} \widehat{\mathcal{L}}_{\mathcal{H}_{\mathrm{all}}}(\mathrm{TF})$ and population minima

$$\mathrm{TF}^* = \arg\min_{\mathrm{TF}\in\mathrm{Alg}} \mathcal{L}_{\mathrm{MTL}}(\mathrm{TF})$$

In the following discussion, we drop the subscripts MTL and $\mathcal{H}_{\mathrm{all}}$. The excess MTL risk is decomposed as follows:

$$\begin{aligned}
\mathcal{R}_{\mathrm{MTL}}(\widehat{\mathrm{TF}}) &= \mathcal{L}(\widehat{\mathrm{TF}}) - \mathcal{L}(\mathrm{TF}^*) \\
&= \underbrace{\mathcal{L}(\widehat{\mathrm{TF}}) - \widehat{\mathcal{L}}(\widehat{\mathrm{TF}})}_{a} + \underbrace{\widehat{\mathcal{L}}(\widehat{\mathrm{TF}}) - \widehat{\mathcal{L}}(\mathrm{TF}^*)}_{b} + \underbrace{\widehat{\mathcal{L}}(\mathrm{TF}^*) - \mathcal{L}(\mathrm{TF}^*)}_{c}.
\end{aligned}$$

Since $\widehat{\mathrm{TF}}$ is the minimizer of empirical risk, we have $b \le 0$.

**Step 1: (Concentration bound $|\mathcal{L}(\mathrm{TF}) - \widehat{\mathcal{L}}(\mathrm{TF})|$ for a fixed $\mathrm{TF} \in \mathrm{Alg}$)** Define the test/train risks of each task as follows:

$$\widehat{\mathcal{L}}_m(\mathrm{TF}) := \frac{1}{n}\sum_{t=1}^{n} \ell\left(r_{mt}(a_{mt}), \mathrm{TF}\left(\widehat{r}_{mt}(a_{mt})|\mathcal{H}_m^{t-1}, a_{mt}\right)\right), \quad \text{and}$$

$$\mathcal{L}_m(\mathrm{TF}) := \mathbb{E}_{\mathcal{H}_m}\left[\widehat{\mathcal{L}}_m(\mathrm{TF})\right] = \mathbb{E}_{\mathcal{H}_m}\left[\frac{1}{n}\sum_{t=1}^{n}\ell\left(r_{mt}(a_{mt}), \mathrm{TF}\left(\widehat{r}_{mt}(a_{mt})|\mathcal{H}_m^{t-1}, a_{mt}\right)\right)\right], \quad \forall m \in [M].$$

Define the random variables $X_{m,t} = \mathbb{E}\left[\widehat{\mathcal{L}}_t(\mathrm{TF}) \mid \mathcal{H}_m^t\right]$ for $t \in [n]$ and $m \in [M]$, that is, $X_{m,t}$ is the expectation over $\widehat{\mathcal{L}}_t(\mathrm{TF})$ given training sequence $\mathcal{H}_m^t = \{(a_{mt'}, r_{mt'})\}_{t'=1}^{t}$ (which are the filtrations). With this, we have that $X_{m,n} = \mathbb{E}\left[\widehat{\mathcal{L}}_m(\mathrm{TF}) \mid \mathcal{H}_m^n\right] = \widehat{\mathcal{L}}_m(\mathrm{TF})$ and $X_{m,0} = \mathbb{E}\left[\widehat{\mathcal{L}}_m(\mathrm{TF})\right] = \mathcal{L}_m(\mathrm{TF})$. More generally, $(X_{m,0}, X_{m,1}, \ldots, X_{m,n})$ is a martingale sequence since, for every $m \in [M]$, we have that $\mathbb{E}\left[X_{m,i} \mid \mathcal{H}_m^{t-1}\right] = X_{m,t-1}$. For notational simplicity, in the following discussion, we omit the subscript $m$ from $a, r$ and $\mathcal{H}$ as they will be clear from the left-hand-side variable $X_{m,t}$. We have that

$$\begin{aligned}
X_{m,t} &= \mathbb{E}\left[\frac{1}{n}\sum_{t=1}^{n}\ell\left(r_{t'}, \mathrm{TF}\left(\widehat{r}_{t'}|\mathcal{H}^{t'-1}, a_{t'}\right)\right)\middle| \mathcal{H}^t\right] \\
&= \frac{1}{n}\sum_{t'=1}^{t}\ell\left(r_{t'}, \mathrm{TF}\left(\widehat{r}_{t'}|\mathcal{H}^{t'-1}, a_{t'}\right)\right) + \frac{1}{n}\sum_{t'=t+1}^{n}\mathbb{E}\left[\ell\left(r_{t'}, \mathrm{TF}\left(\widehat{r}_{t'}|\mathcal{H}^{t'-1}, a_{t'}\right)\right) \mid \mathcal{H}^t\right]
\end{aligned}$$

Using the similar steps as in Li et al. (2023) we can show that

$$|X_{m,t} - X_{m,t-1}| \overset{(a)}{\le} \frac{B}{n} + \sum_{t'=t+1}^{n}\frac{K}{t'n} \le \frac{B + K\log n}{n}.$$

where, $(a)$ follows by using the fact that loss function $\ell(\cdot, \cdot)$ is bounded by $B$, and error stability assumption.

Recall that $\left|\mathcal{L}_m(\mathrm{TF}) - \widehat{\mathcal{L}}_m(\mathrm{TF})\right| = |X_{m,0} - X_{m,n}|$ and for every $m \in [M]$, we have $\sum_{t=1}^{n}|X_{m,t} - X_{m,t-1}|^2 \le \frac{(B+K\log n)^2}{n}$. As a result, applying Azuma-Hoeffding's inequality, we obtain

$$\mathbb{P}\left(\left|\mathcal{L}_m(\mathrm{TF}) - \widehat{\mathcal{L}}_m(\mathrm{TF})\right| \ge \tau\right) \le 2e^{-\frac{n\tau^2}{2(B+K\log n)^2}}, \quad \forall m \in [M] \tag{10}$$

Let us consider $Y_m := \mathcal{L}_m(\mathrm{TF}) - \widehat{\mathcal{L}}_m(\mathrm{TF})$ for $m \in [M]$. Then, $(Y_m)_{m=1}^{M}$ are i.i.d. zero mean sub-Gaussian random variables. There exists an absolute constant $c_1 > 0$ such that, the subgaussian

norm, denoted by $\|\cdot\|_{\psi_2}$, obeys $\|Y_m\|_{\psi_2}^2 < \frac{c_1(B+K\log n)^2}{n}$ via Proposition 2.5.2 of (Vershynin, 2018). Applying Hoeffding's inequality, we derive

$$\mathbb{P}\left(\left|\frac{1}{M}\sum_{m=1}^M Y_t\right| \geq \tau\right) \leq 2e^{-\frac{cnM\tau^2}{(B+K\log n)^2}} \implies \mathbb{P}(|\widehat{\mathcal{L}}(\mathrm{TF}) - \mathcal{L}(\mathrm{TF})| \geq \tau) \leq 2e^{-\frac{cnM\tau^2}{(B+K\log n)^2}}$$

where $c > 0$ is an absolute constant. Therefore, we have that for any $\mathrm{TF} \in \mathrm{Alg}$, with probability at least $1 - 2\delta$,

$$|\widehat{\mathcal{L}}(\mathrm{TF}) - \mathcal{L}(\mathrm{TF})| \leq (B + K\log n)\sqrt{\frac{\log(1/\delta)}{cnM}} \tag{11}$$

**Step 2: (Bound $\sup_{\mathrm{TF}\in\mathrm{Alg}} |\mathcal{L}(\mathrm{TF}) - \widehat{\mathcal{L}}(\mathrm{TF})|$ where $\mathrm{Alg}$ is assumed to be a continuous search space)**. Let

$$h(\mathrm{TF}) := \mathcal{L}(\mathrm{TF}) - \widehat{\mathcal{L}}(\mathrm{TF})$$

and we aim to bound $\sup_{\mathrm{TF}\in\mathrm{Alg}} |h(\mathrm{TF})|$. Following Definition C.3, for $\varepsilon > 0$, let $\mathrm{Alg}_\varepsilon$ be a minimal $\varepsilon$-cover of $\mathrm{Alg}$ in terms of distance metric $\rho$. Therefore, $\mathrm{Alg}_\varepsilon$ is a discrete set with cardinality $|\mathrm{Alg}_\varepsilon| := \mathcal{N}(\mathrm{Alg}, \rho, \varepsilon)$. Then, we have

$$\sup_{\mathrm{TF}\in\mathrm{Alg}} |\mathcal{L}(\mathrm{TF}) - \widehat{\mathcal{L}}(\mathrm{TF})| \leq \sup_{\mathrm{TF}\in\mathrm{Alg}'}\min_{\mathrm{TF}\in\mathrm{Alg}_\varepsilon} |h(\mathrm{TF}) - h(\mathrm{TF}')| + \max_{\mathrm{TF}\in\mathrm{Alg}_\varepsilon} |h(\mathrm{TF})|.$$

We will first bound the quantity $\sup_{\mathrm{TF}\in\mathrm{Alg}'}\min_{\mathrm{TF}\in\mathrm{Alg}_\varepsilon} |h(\mathrm{TF}) - h(\mathrm{TF}')|$. We will utilize that loss function $\ell(\cdot,\cdot)$ is $C$-Lipschitz. For any $\mathrm{TF} \in \mathrm{Alg}$, let $\mathrm{TF} \in \mathrm{Alg}_\varepsilon$ be its neighbor following Definition C.3. Then we can show that

$$\left|\widehat{\mathcal{L}}(\mathrm{TF}) - \widehat{\mathcal{L}}(\mathrm{TF}')\right|$$

$$= \left|\frac{1}{nM}\sum_{m=1}^M\sum_{t=1}^n \left(\ell\left(r_{mt}(a_{mt}), \mathrm{TF}\left(\widehat{r}_{mt}(a_{mt})|\mathcal{H}_m^{t-1}, a_{mt}\right)\right) - \ell\left(r_{mt}(a_{mt}), \mathrm{TF}'\left(\widehat{r}_{mt}(a_{mt})|\mathcal{H}_m^{t-1}, a_{mt}\right)\right)\right)\right|$$

$$\leq \frac{L}{nM}\sum_{m=1}^M\sum_{t=1}^n \left\|\mathrm{TF}\left(\widehat{r}_{mt}(a_{mt})|\mathcal{H}_m^{t-1}, a_{mt}\right) - \mathrm{TF}'\left(\widehat{r}_{mt}(a_{mt})|\mathcal{H}_m^{t-1}, a_{mt}\right)\right\|_{\ell_2}$$

$$\leq L\varepsilon.$$

Note that the above bound applies to all data-sequences, we also obtain that for any $\mathrm{TF} \in \mathrm{Alg}$,

$$\left|\mathcal{L}(\mathrm{TF}) - \mathcal{L}(\mathrm{TF}')\right| \leq L\varepsilon.$$

Therefore we can show that,

$$\sup_{\mathrm{TF}\in\mathrm{Alg}}\min_{\mathrm{TF}} \in \mathrm{Alg}_\varepsilon |h(\mathrm{TF}) - h(\mathrm{TF}F')|$$

$$\leq \sup_{\mathrm{TF}\in\mathrm{Alg}}\min_{\mathrm{TF}} \in \mathrm{Alg}_\varepsilon \left|\widehat{\mathcal{L}}(\mathrm{TF}) - \widehat{\mathcal{L}}(\mathrm{TF}')\right| + \left|\mathcal{L}(\mathrm{TF}) - \mathcal{L}(\mathrm{TF}')\right| \leq 2L\varepsilon. \tag{12}$$

Next we bound the second term $\max_{\mathrm{TF}\in\mathrm{Alg}_\varepsilon} |h(\mathrm{TF})|$. Applying union bound directly on $\mathrm{Alg}_\varepsilon$ and combining it with equation 11, then we will have that with probability at least $1 - 2\delta$,

$$\max_{\mathrm{TF}\in\mathrm{Alg}_\varepsilon} |h(\mathrm{TF})| \leq (B + K\log n)\sqrt{\frac{\log(\mathcal{N}(\mathrm{Alg}, \rho, \varepsilon)/\delta)}{cnM}}$$

Combining the upper bound above with the perturbation bound equation 12, we obtain that

$$\max_{\mathrm{TF}\in\mathrm{Alg}} |h(\mathrm{TF})| \leq 2C\varepsilon + (B + K\log n)\sqrt{\frac{\log(\mathcal{N}(\mathrm{Alg}, \rho, \varepsilon)/\delta)}{cnM}}.$$

It follows then that

$$\mathcal{R}_{\mathrm{MTL}}(\widehat{\mathrm{TF}}) \leq 2\sup_{\mathrm{TF}\in\mathrm{Alg}} |\mathcal{L}(\mathrm{TF}) - \widehat{\mathcal{L}}(\mathrm{TF})| \leq 4C\varepsilon + 2(B + K\log n)\sqrt{\frac{\log(\mathcal{N}(\mathrm{Alg}, \rho, \varepsilon)/\delta)}{cnM}}$$

Again by setting $\varepsilon = 1/\sqrt{nM}$

$$\mathcal{L}(\widehat{\mathrm{TF}}) - \mathcal{L}(\mathrm{TF}^*) \leq \frac{4C}{\sqrt{nM}} + 2(B + K \log n)\sqrt{\frac{\log(\mathcal{N}(\mathrm{Alg}, \rho, \varepsilon)/\delta)}{cnM}}$$

The claim of the theorem follows. $\square$

**Definition C.2.** (Covering number) Let $Q$ be any hypothesis set and $d(q, q') \geq 0$ be a distance metric over $q, q' \in \mathcal{Q}$. Then, $\tilde{Q} = \{q_1, \ldots, q_N\}$ is an $\varepsilon$-cover of $Q$ with respect to $d(\cdot, \cdot)$ if for any $q \in \mathcal{Q}$, there exists $q_i \in \tilde{Q}$ such that $d(q, q_i) \leq \varepsilon$. The $\varepsilon$-covering number $\mathcal{N}(Q, d, \varepsilon)$ is the cardinality of the minimal $\varepsilon$-cover.

**Definition C.3.** (Algorithm distance). Let Alg be an algorithm hypothesis set and $\mathcal{H} = (a_t, r_t)_{t=1}^n$ be a sequence that is admissible for some task $m \in [M]$. For any pair $\mathrm{TF}, \mathrm{TF}' \in \mathrm{Alg}$, define the distance metric $\rho(\mathrm{TF}, \mathrm{TF}') := \sup_{\mathcal{H}} \frac{1}{n} \sum_{t=1}^n \left\| \mathrm{TF}(\hat{r}_t | \mathcal{H}^{t-1}, a_t) - \mathrm{TF}'(\hat{r}_t | \mathcal{H}^{t-1}, a_t) \right\|_{\ell_2}$.

*Remark* C.4. **(Stability Factor)** The work of Li et al. (2023) also characterizes the stability factor $K$ in Assumption 7.1 with respect to the transformer architecture. Assuming loss $\ell(\cdot, \cdot)$ is C-Lipschitz, the algorithm induced by $\mathrm{TF}(\cdot)$ obeys the stability assumption with $K = 2C\left((1 + \Gamma)e^\Gamma\right)^L$, where the norm of the transformer weights are upper bounded by $O(\Gamma)$ and there are $L$-layers of the transformer.

*Remark* C.5. **(Covering Number)** From Lemma 16 of Lin et al. (2023) we have the following upper bound on the covering number of the transformer class $\mathrm{TF}_\Theta$ as

$$\log(\mathcal{N}(\mathrm{Alg}, \rho, \varepsilon)) \leq O(L^2 D^2 J)$$

where $L$ is the total number of layers of the transformer and $J$ and, $D$ denote the upper bound to the number of heads and hidden neurons in all the layers respectively. Note that this covering number holds for the specific class of transformer architecture discussed in section 2 of (Lin et al., 2023).

## C.2 GENERALIZATION ERROR TO NEW TASK

**Theorem C.6.** *(Transfer Risk) Consider the setting of Theorem 7.2 and assume the source tasks are independently drawn from task distribution $\mathcal{T}$. Let $\widehat{\mathrm{TF}}$ be the empirical solution of (ERM) and $g \sim \mathcal{T}$. Then with probability at least $1 - 2\delta$, the expected excess transfer learning risk is bounded by*

$$\mathbb{E}_g\left[\mathcal{R}_g(\widehat{\mathrm{TF}})\right] \leq 4\frac{C}{\sqrt{M}} + B\sqrt{\frac{2\log(\mathcal{N}(\mathrm{Alg}, \rho, \varepsilon)/\delta)}{M}}$$

*where, $\mathcal{N}(\mathrm{Alg}, \rho, \varepsilon)$ is the covering number of transformer $\widehat{\mathrm{TF}}$.*

*Proof.* Let the target task $g$ be sampled from $\mathcal{T}$, and the test set $\mathcal{H}_g = \{a_t, r_t\}_{t=1}^n$. Define empirical and population risks on $g$ as $\widehat{\mathcal{L}}_g(\mathrm{TF}) = \frac{1}{n} \sum_{t=1}^n \ell\left(r_t(a_{mt}), \mathrm{TF}(\hat{r}_t(a_{mt}) | \mathcal{H}_g^{t-1}, a_t)\right)$ and $\mathcal{L}_g(\mathrm{TF}) = \mathbb{E}_{\mathcal{H}_g}\left[\widehat{\mathcal{L}}_g(\mathrm{TF})\right]$. Again we drop $\Theta$ from the transformer notation. Then the expected excess transfer risk following (ERM) is defined as

$$\mathbb{E}_g\left[\mathcal{R}_g(\widehat{\mathrm{TF}})\right] = \mathbb{E}_{\mathcal{H}_g}\left[\mathcal{L}_g(\widehat{\mathrm{TF}})\right] - \arg\min_{\mathrm{TF} \in \mathrm{Alg}} \mathbb{E}_{\mathcal{H}_g}\left[\mathcal{L}_g(\mathrm{TF})\right]. \tag{13}$$

where $\mathcal{A}$ is the set of all algorithms. The goal is to show a bound like this

$$\mathbb{E}_g\left[\mathcal{R}_g(\widehat{\mathrm{TF}})\right] \leq \min_{\varepsilon \geq 0}\left\{4C\varepsilon + B\sqrt{\frac{2\log(\mathcal{N}(\mathrm{Alg}, \rho, \varepsilon)/\delta)}{T}}\right\}$$

where $\mathcal{N}(\mathrm{Alg}, \rho, \varepsilon)$ is the covering number.

**Step 1 ((Decomposition):** Let $\mathrm{TF}^* = \arg\min_{\mathrm{TF} \in \mathrm{Alg}} \mathbb{E}_g\left[\mathcal{L}_g(\mathrm{TF})\right]$. The expected transfer learning excess test risk of given algorithm $\widehat{\mathrm{TF}} \in \mathrm{Alg}$ is formulated as

$$\widehat{\mathcal{L}}_m(\mathrm{TF}) := \frac{1}{n} \sum_{t=1}^n \ell\left(r_{mt}(a_{mt}), \mathrm{TF}(\hat{r}_{mt}(a_{mt}) | \mathcal{D}_m^{t-1}, a_{mt})\right), \quad \text{and}$$

$$\mathcal{L}_m(\mathrm{TF}) := \mathbb{E}_{\mathcal{H}_m}\left[\widehat{\mathcal{L}}_t(\mathrm{TF})\right] = \mathbb{E}_{\mathcal{H}_m}\left[\frac{1}{n} \sum_{t=1}^n \ell\left(r_{mt}(a_{mt}), \mathrm{TF}(\hat{r}_{mt}(a_{mt}) | \mathcal{D}_m^{t-1}, a_{mt})\right)\right], \quad \forall m \in [M].$$

Then we can decompose the risk as

$$\mathbb{E}_g\left[\mathcal{R}_g(\widehat{\mathrm{TF}})\right] = \mathbb{E}_g\left[\mathcal{L}_g(\widehat{\mathrm{TF}})\right] - \mathbb{E}_g\left[\mathcal{L}_g\left(\mathrm{TF}^*\right)\right]$$

$$= \underbrace{\mathbb{E}_g\left[\mathcal{L}_g(\widehat{\mathrm{TF}})\right] - \widehat{\mathcal{L}}_{\mathcal{H}_{\mathrm{all}}}(\widehat{\mathrm{TF}})}_{a} + \underbrace{\widehat{\mathcal{L}}_{\mathcal{H}_{\mathrm{all}}}(\widehat{\mathrm{TF}}) - \widehat{\mathcal{L}}_{\mathcal{H}_{\mathrm{all}}}\left(\mathrm{TF}^*\right)}_{b} + \underbrace{\widehat{\mathcal{L}}_{\mathcal{H}_{\mathrm{all}}}\left(\mathrm{TF}^*\right) - \mathbb{E}_g\left[\mathcal{L}_g\left(\mathrm{TF}^*\right)\right]}_{c}.$$

Here since $\widehat{\mathrm{TF}}$ is the minimizer of training risk, $b < 0$. Then we obtain

$$\mathbb{E}_g\left[\mathcal{R}_g(\widehat{\mathrm{TF}})\right] \leq 2 \sup_{\mathrm{TF}\in\mathrm{Alg}} \left|\mathbb{E}_g\left[\mathcal{L}_g(\mathrm{TF})\right] - \frac{1}{M}\sum_{m=1}^{M}\widehat{\mathcal{L}}_m(\mathrm{TF})\right|. \tag{14}$$

**Step 2 (Bounding equation 14)** For any $\mathrm{TF} \in \mathrm{Alg}$, let $X_t = \widehat{\mathcal{L}}_t(\mathrm{TF})$ and we observe that

$$\mathbb{E}_{m\sim\mathcal{T}}\left[X_t\right] = \mathbb{E}_{m\sim\mathcal{T}}\left[\widehat{\mathcal{L}}_m(\mathrm{TF})\right] = \mathbb{E}_{m\sim\mathcal{T}}\left[\mathcal{L}_m(\mathrm{TF})\right] = \mathbb{E}_g\left[\mathcal{L}_g(\mathrm{TF})\right]$$

Since $X_m, m \in [M]$ are independent, and $0 \leq X_m \leq B$, applying Hoeffding's inequality obeys

$$\mathbb{P}\left(\left|\mathbb{E}_g\left[\mathcal{L}_g(\mathrm{TF})\right] - \frac{1}{M}\sum_{m=1}^{M}\widehat{\mathcal{L}}_m(\mathrm{TF})\right| \geq \tau\right) \leq 2e^{-\frac{2M\tau^2}{B^2}}.$$

Then with probability at least $1 - 2\delta$, we have that for any $\mathrm{TF} \in \mathrm{Alg}$,

$$\left|\mathbb{E}_g\left[\mathcal{L}_g(\mathrm{TF})\right] - \frac{1}{M}\sum_{m=1}^{M}\widehat{\mathcal{L}}_m(\mathrm{TF})\right| \leq B\sqrt{\frac{\log(1/\delta)}{2M}}. \tag{15}$$

Next, let $\mathrm{Alg}_\varepsilon$ be the minimal $\varepsilon$-cover of $\mathrm{Alg}$ following Definition C.2, which implies that for any task $g \sim \mathcal{T}$, and any $\mathrm{TF} \in \mathrm{Alg}$, there exists $\mathrm{TF}' \in \mathrm{Alg}_\varepsilon$

$$\left|\mathcal{L}_g(\mathrm{TF}) - \mathcal{L}_g\left(\mathrm{TF}'\right)\right|, \left|\widehat{\mathcal{L}}_g(\mathrm{TF}) - \widehat{\mathcal{L}}_g\left(\mathrm{TF}'\right)\right| \leq C\varepsilon. \tag{16}$$

Since the distance metric following Definition 3.4 is defined by the worst-case datasets, then there exists $\mathrm{TF}' \in \mathrm{Alg}_\varepsilon$ such that

$$\left|\mathbb{E}_g\left[\mathcal{L}_g(\mathrm{TF})\right] - \frac{1}{M}\sum_{m=1}^{M}\widehat{\mathcal{L}}_m(\mathrm{TF})\right| \leq 2C\varepsilon.$$

Let $\mathcal{N}(\mathrm{Alg}, \rho, \varepsilon) = |\mathrm{Alg}_\varepsilon|$ be the $\varepsilon$-covering number. Combining the above inequalities (equation 14, equation 15, and equation 16), and applying union bound, we have that with probability at least $1 - 2\delta$,

$$\mathbb{E}_g\left[\mathcal{R}_g(\widehat{\mathrm{TF}})\right] \leq \min_{\varepsilon\geq 0}\left\{4C\varepsilon + B\sqrt{\frac{2\log(\mathcal{N}(\mathrm{Alg}, \rho, \varepsilon)/\delta)}{M}}\right\}$$

Again by setting $\varepsilon = 1/\sqrt{M}$

$$\mathcal{L}(\widehat{\mathrm{TF}}) - \mathcal{L}\left(\mathrm{TF}^*\right) \leq \frac{4C}{\sqrt{M}} + 2B\sqrt{\frac{\log(\mathcal{N}(\mathrm{Alg}, \rho, \varepsilon)/\delta)}{cM}}$$

The claim of the theorem follows. $\qquad\qquad\square$

*Remark* C.7. (Dependence on $n$)  In this remark, we briefly discuss why the expected excess risk for target task $\mathcal{T}$ does not depend on samples $n$. The work of Li et al. (2023) pointed out that the MTL pretraining process identifies a favorable algorithm that lies in the span of the $M$ source tasks. This is termed as inductive bias (see section 4 of Li et al. (2023)) (Soudry et al., 2018; Neyshabur et al., 2017). Such bias would explain the lack of dependence of the expected excess transfer risk on $n$ during transfer learning. This is because given a target task $g \sim \mathcal{T}$, the TF can use the learnt favorable algorithm to conduct a discrete search over span of the $M$ source tasks and return the source task that best fits the new target task. Due to the discrete search space over the span of $M$ source tasks, it is not hard to see that, we need $n \propto \log(M)$ samples (which is guaranteed by the $M$ source tasks) rather than $n \propto d$ (for the linear setting).

## C.3 TABLE OF NOTATIONS

| Notations | Definition |
|---|---|
| $M$ | Total number of tasks |
| $d$ | Dimension of the feature |
| $\mathcal{A}_m$ | Action set of the $m$-th task |
| $\mathcal{X}_m$ | Feature space of $m$-th task |
| $M_{\text{test}}$ | Tasks for testing |
| $M_{\text{pre}}$ | Total Tasks for pretraining |
| $\mathbf{x}(m,a)$ | Feature of action $a$ in task $m$ |
| $\boldsymbol{\theta}_{m,*}$ | Hidden parameter for the task $m$ |
| $\mathcal{T}_{\text{pre}}$ | Pretraning distribution on tasks |
| $\mathcal{T}_{\text{test}}$ | Testing distribution on tasks |
| $n$ | Total horizon for each task $m$ |
| $\mathcal{H}_m = \{I_t, r_t\}_{t=1}^n$ | Dataset sampled for the $m$-th task containing $n$ samples |
| $\mathcal{H}_m^t = \{I_s, r_s\}_{s=1}^t$ | Dataset sampled for the $m$-th task containing samples from round $s = 1$ to $t$ |
| $\mathbf{w}$ | Transformer model parameter |
| $\text{TF}_{\mathbf{w}}$ | Transformer with model parameter $\mathbf{w}$ |
| $\mathcal{D}_{\text{pre}}$ | Pretraining in-context distribution |
| $\mathcal{H}_{\text{train}}$ | Training in-context dataset |
| $\mathcal{D}_{\text{test}}$ | Testing in-context distribution |

Table 1: Table of Notations

