# OpenReview forum: "Pretraining Decision Transformers with Reward Prediction for In-Context Structured Bandit Learning"
_ICLR.cc/2025/Conference — Submitted to ICLR 2025_

### Official Review · Reviewer_5yut · 2024-11-03

**Soundness:** 2
**Presentation:** 2
**Contribution:** 2
**Rating:** 3
**Confidence:** 4

**Summary:**

This paper employs a Transformer model for decision-making in multi-armed bandit problems. Rather than generating actions directly, the authors propose training the Transformer to predict the reward for each arm. Actions are then selected greedily, with occasional random exploration, based on the reward estimates. The Transformer is trained on trajectories from a near-optimal expert, such as standard UCB or Thompson sampling algorithms. Results demonstrate that the trained Transformer outperforms traditional learning algorithms across a variety of bandit problems with shared structural features. The authors also present theoretical results quantifying generalization and transfer errors in relation to the training loss.

**Strengths:**

The paper introduces an alternative pre-training method for simple decision-making environments, such as bandits. In my opinion, this design is well-suited for the bandit context, and the algorithm performs effectively in both synthetic and real environments. The authors also provide ample ablation studies to explore different setups and gain insight into the conditions under which this algorithm performs well. In general, I find no significant issues regarding the originality or clarity of this paper.

**Weaknesses:**

I think the biggest weakness of the paper is that it fails to provide sufficient understanding of why this algorithm works well—specifically, it lacks insights or supporting evidence on why this design would outperform other algorithms and what key takeaways it offers for theoretical researchers and practitioners. For example, I find that the experimental section and the theory section feel somewhat disconnected. The theoretical results focus on generalization errors and transferability with respect to the pre-training loss. However, this pre-training loss does not directly correlate with the regret, which is the focus of the empirical study. This paper feels like a mixture of observations and results on different aspects of the problem, instead of a systematic effort to understand this pre-training pipeline.

**Questions:**

1. Could you provide a systematic explanation of why this design benefits decision-making, particularly in the shared-structure bandit setting? After reading the paper, I am unclear about the authors' motivation for studying the bandit problem with a shared structure. In my opinion, there is a lack of insight into why a Transformer-based decision-maker would succeed even in a simple bandit environment.

2. While I understand that the proposed algorithm does not rely on the optimal action, it would be beneficial to include a comparison with the optimal action to assess the performance difference. This addition could help clarify how much of the performance improvement is attributable to the optimal decision in the training data itself. Furthermore, I believe the paper should cite the following related work, which is closely related.

Supervised Pretraining Can Learn In-Context Reinforcement Learning, Jonathan N. Lee, Annie Xie, Aldo Pacchiano, Yash Chandak, Chelsea Finn, Ofir Nachum, Emma Brunskill.

---

> ### Author Response · Authors · 2024-11-19
> **Response to Reviewer 5yut**
>
> We thank the reviewer for their reviews. There are several misunderstandings of the reviewer which we wish to clarify.
>
> Weakness: We explain in detail why PreDeToR works well in Appendix A.11. We also refer to it in the main paper (see line 383). We analyze it through the tasks where the optimal action is 5. We see that PreDeToR explores in two phases during testing. In the first phase for the first few rounds, it recommends actions based on the strong prior observed from the training data. This includes suggesting the optimal actions 2,3,5,6,7,10 which are the optimal actions across all tasks from the task distribution. In the second phase after observing a few in-context examples from the test task (in the graph we plot the action samples from the test tasks where optimal action is 5) it starts suggesting the optimal action to the task.
>
> Questions:
>
> 1. The main goal of our paper is to understand how the PreDeToR leverages shared latent structure across tasks. The shared structure lets the transformer learn a correlation between tasks, the action selected, and the underlying feedback. The [1] and [2] papers are few examples that exactly study the same setting as ours.
>
>     [1] fails in settings where the number of interactions is less because it needs the optimal action or a greedy approximation of it. In contrast, PreDeToR predicts the next reward of each action and we show that this helps in learning a latent representation that is generalizable to unseen tasks. This is best illustrated with this simple example. Consider two users: User 1, who prefers item 1 over item 2, and User 2 who prefers item 2 over item 1. We have an interaction history of many such users. Then we have a new user 3 with no past interaction history. After observing a few in-context examples/feedback the transformer is able to figure out which item to suggest to the user. We discuss this exploration in detail in Appendix A.11 and also extend the exploration study to unseen items in Appendix A.12.
>
> 2. In the linear bandit setting we show how our algorithm matches the oracle LinUCB which knows the underlying linear feedback structure. The DPT with access to the optimal action achieves the same performance (see the paper [1]) against LinUCB. However, we outperform DPT greedy which approximates the optimal action.
>
> 3. We have cited [1] many times in our paper. This is the DPT paper and our main competitor.
>
> [1] Supervised Pretraining Can Learn In-Context Reinforcement Learning, Jonathan N. Lee, Annie Xie, Aldo Pacchiano, Yash Chandak, Chelsea Finn, Ofir Nachum, Emma Brunskill.
> [2] Transformers as Statisticians: Provable In-Context Learning with In-Context Algorithm Selection, Yu Bai, Fan Chen, Huan Wang, Caiming Xiong, Song Mei

---

> > ### Comment · Reviewer_5yut · 2024-11-23
> > **Reply to author**
> >
> > Thanks for replying.
> >
> > First, please let me explain why I mistakenly claimed that the paper does not cite the DPT paper. I was looking at the experiment section and did not find the mention of DPT (which I think would be helpful for better understanding the proposed PreDeToR algorithm). Then I searched for the name of the DPT paper, but it turned out that the search functionality in my PDF reader failed. I apology for this.
> >
> > After reading the reply, I think my concerns are only partially addressed. Please see my response to the three replies from the author correspondingly:
> >
> > First, I do not think the explanation in Appendix 11 is comprehensive enough to explain the advantage, since it is only performed on one setting and there are no theoretical insights into why it behaves this way. It would be helpful to add more ablation studies to highlight the essential parts that make this algorithm successful.
> >
> > Second, I can see what the examples on users 1, 2, and 3 are suggesting. However, the trouble I have with this kind of intuition is that I feel there is a missing connecting element in the explanation. If the authors claim that the algorithm is learning the latent elements, then it would be helpful to provide a formal definition or some theoretical insights into these latent factors. If that is not possible, at the very least, empirical results should be provided to convince us that the algorithm is indeed learning the latent factors. As I recall, this work only shows results in terms of regret.
> >
> > Third, I understand that DPT might not be the best-performing algorithm. What I am suggesting is that when training the PreDeToR algorithm, you could also use the optimal decision for pre-training. This alone would help us better understand the algorithm.
> >
> > Please let me know if you have further questions. So far, if there were an option of giving a score of 4, I would choose 4, but only 3 and 5 are available.

---

> > > ### Author Response · Authors · 2024-11-23
> > > **Response to Reviewer 5yut**
> > >
> > > Thanks for engaging with us. We address the issue raised by the reviewer and clear some misunderstandings:
> > >
> > > **How to assess PreDeToR learning latent representation?**: The way we assessed it is that during training time (as well as testing time) we do not pass in the feature information of the selected action to PreDeToR. Now consider the linear bandit setting. The optimal algorithm LinUCB uses the feature information and the linear structure assumption to reduce the cumulative regret optimally. Please note that the goal of reducing cumulative regret *optimally* involves utilizing the feature information. However, PreDeToR does not have access to feature information (or the linear feedback assumption) and matches the performance of LinUCB for all test tasks. This naturally leads to the conclusion that  PreDeToR is able to learn a latent representation.
> > >
> > > **How do DPT and other Deep RL works show learning representation?**: The DPT paper [1] uses the same argument as above to argue the DPT is learning the latent representation which is clearly stated in their section "Leveraging structure from suboptimal data". Again we note that the goal of minimizing the cumulative regret is linked to learning the latent structure.
> > >
> > > **Why only a linear setting to understand learning representation in Appendix 11?** We use only a linear setting to understand that PreDeToR is a learning representation because the optimal algorithm (LinUCB) in this setting is known to us. Note that LinUCB is an oracle algorithm that leverages structure and the linear feedback assumption to minimize cumulative regret. Since PreDeToR matches the performance of LinUCB in minimizing cumulative regret we conclude that PreDeToR (with no access to feature information or feedback structure) is learning a latent representation. This is a standard empirical approach to assess whether the learning algorithm is leveraging latent structure (see [1]). Again note that *the goal of minimizing the cumulative regret is linked to learning the latent structure.*
> > >
> > > **Supporting theory**: Our theoretical analysis is actually based on the observation of the exploration stated in Appendix 11. In Appendix 11 we show that PreDeToR conducts a two-phase exploration: First phase is motivated by its prior over training data, and the second phase is motivated by in-context data from the test task. Moreover, PreDeToR has a low prediction error over the optimal action in the test task. This necessitates an analysis based on empirical risk minimization where we show that PreDeToR is able to identify the test task because of its training on the train tasks.
> > >
> > > **More ablation study to understand advantage**: This is a very general comment and we do not follow it. Can the reviewer be more specific about what type of ablation experiments they are seeking? We showed in linear, non-linear, changing feature bandits, Movielens dataset, Yelp dataset, bilinear bandits, and low-rank bandits that we outperform other baselines. Again, to asses why we only use linear bandit setting to understand exploration and representation learning please refer to the above answer. If the reviewer has some other approach in mind then please let us know.
> > >
> > > **Training PreDeToR with optimal arm**: We are not sure what the reviewer is referring to. Currently, PreDeToR minimizes the MSE between the observed reward of the next action and the predicted reward for that action. Let's say PreDeToR has additional access to the optimal action for that training task. However, if we only train the PreDeToR now by minimizing the MSE of the reward of the optimal action and the predicted reward of the optimal action, then PreDeToR will also learn the latent structure given the different tasks have different optimal actions (and the tasks share the same shared structure). However, note that such an oracle PreDeToR is very similar to DPT. Hope this clears your doubts.
> > >
> > > [1] Supervised Pretraining Can Learn In-Context Reinforcement Learning, Jonathan N. Lee, Annie Xie, Aldo Pacchiano, Yash Chandak, Chelsea Finn, Ofir Nachum, Emma Brunskill.

---

> ### Comment · Area_Chair_E1CA · 2024-11-22
> **Please engage with authors**
>
> Dear Reviewer,
>
> Please engage with the authors. I've seen that you ask the authors to cite a related work which they cite multiple times in their work. This is not inspiring confidence on whether a careful assessment of this work is being conducted. I hope this is resolved.
>
> Thanks!
>
> The AC

---

> ### Comment · Reviewer_5yut · 2024-11-27
> **Reply to the author**
>
> Thanks for the reply.
>
> Q: How to assess PreDeToR learning latent representation?
>
> R: Now I understand why it learn a latent structure.
>
> Q: Why only a linear setting to understand learning representation in Appendix 11
>
> R: This is because I feel (it could be subjective) it is more natural to start from stochastic bandit to see the effect first, analyze why this algorithm is working, and what make this work; then move to the linear bandit setting, which is more complex. We could have a better understanding on the linear setting based on what we have observed on the stochastic bandit setting.
>
> Q: Supporting theory
>
> R: Can you direct me to the experimental part where you mention "PreDeToR has a low prediction error"? I can only find similar results in figure 2 c, and the majority of the experimental story is based on regret.
>
> Q: More ablation study to understand advantage
>
> R: For example, I can suggest two approaches, although you might not agree with me. 1) I think the first step could be taking the linear structure out (for example, stochastic bandit), analyzing the performance, and then adding the linear structure. 2) The second step can be comparing the performance with the algorithm with the optimal action (similar in DPT). Because I think one of the good part is that this algorithm does not need optimal action, but the paper never explain why. Intuitively, the optimal action would result in better performance, and we want to make sure we understand exactly why part is making this algorithm work. Therefore, this would be another ablation study that would give us more insight.
>
> Q: Training PreDeToR with optimal arm
>
> R: Please see the last reply. I think it is quite important for us to understand what is making this work. At least intuitively, optimal arm should work better if the testing and pre-training environment are not that different, but DPT (maybe implemented differently) performs much worse compared to your algorithm. I think there is a lack of explanation for that.

---

> ### Author Response · Authors · 2024-11-27
> **Response to reviewer 5yut**
>
> We thank the reviewer for engaging with us. We answer below:
>
> **Linear vs K-armed**: The reviewer suggests that we start with K-armed bandits to understand whether PreDeToR is learning the representation or not. This is how we see it. Consider a K-armed bandit setting with K=10 actions and the number of tasks as 100000. In each train task let us fix a set of $K$ arm means randomly between [0,1] and finally ensure that we have 200 test tasks where the arm means are not the same as any of the train tasks. Crucially note that now there is no shared structure among the tasks (and there is no linear structure as well). *This setting is nearly the same as what we experimented in Figure 3d*. Note that in this experiment now we have A=10 actions but every task has 10 random actions selected both during training and testing. So there is no common structure among the tasks. We now see that PreDeToR has linear regret and performs the worst. Therefore for K-armed bandits when actions have no common structure across tasks the *PreDeToR will fail to learn any meaningful representation*.
>
> **Why DPT perform worse?**: There is a misunderstanding by the reviewer on the DPT used in the paper. We explain in lines 269-273 that we use a DPT-greedy version (also mentioned in [1]) in our experiments that do not use the knowledge of the optimal action, but approximates it from the per-task data. Observe that in the experiments the training data per task consists of a small number of rounds forcing the algorithm to leverage the *shared latent structure* to perform optimally in the test task. DPT-greedy fails to approximate the optimal action per task and so performs poorly. We explain this in detail in lines 304-312.
>
> **Ablation study and optimal arm training**: As suggested by the reviewer we ran an additional ablation study in the linear setting. Note that PreDeToR cannot be trained with the knowledge of optimal action as PreDeToR minimizes the MSE between the observed reward of the next action and the predicted reward for that action. Note that the next action selected by the demonstrator may or may not be the optimal action.
>
> Rather we fall back on DPT (oracle) which has access to the optimal action and simply minimizes the cross-entropy loss between the predicted optimal action and true optimal action. The result is shown below. We see that DPT (oracle) matches the performance of LinUCB (as already shown in [1], we also use their codebase) and outperforms both versions of PreDeToR.
>
> | Algorithm  | Regret (n=25) | Regret (n=50) | Regret (n=75) | Regret (n=100) |
> | ----------------- | ----------- | ------------ | ------------ | ------------ |
> | PreDeToR (Ours)      | 5.43  |    9.9          |    13.93          |     17.75      |
> | PreDeToR-$\tau$ (Ours)      | 5.4  |    9.1          |    12.9          |     15.7      |
> | DPT (oracle)  | 5.12      |  8.04        | 10.2 | 12.1 |
> |  DPT (Greedy)  | 7.29  |     10.91         |     14.88         |    19.37       |
> | LinUCB             | 5.08 |     7.39         |     9.42         |     11.98      |
> | Thomp             | 9.56 |     15.30         |     22.5         |     29.92      |
> | AD             | 9.52 |     15.88        |     21.6         |     29.12      |
>
> **PreDeToR has a low prediction error**: Yes we are referring to Figure 2c only. Here we see that PreDeToR almost matches the prediction error of LinUCB for all optimal actions. Note that LinUCB is the optimal algorithm for this linear setting which has access to feature information and the linear structure of the problem.

---

> ### Comment · Reviewer_5yut · 2024-12-01
> **Reply to the Response**
>
> Linear vs K-armed: Thanks for clarifying. I am curious on why the performance on the stochastic setting to be bad, because the whole design mechanism seems to be quite universal. Also, I think there are some subtle difference between a stochastic setting and a linear setting with new actions, the biggest one would be that the variance per arm in the linear setting would be much larger if we ignore the context and treat the linear bandit as stochastic bandit.
>
> Why DPT perform worse: Okay now I understand this. But just like the way you treat linear bandit first instead of stochastic bandit, I always feel there is a part missing such that I don't know the right intuition behind the algorithms. If we want to compare A with B, it is always helpful to vary one factor and keep all others the same. If you let two factors vary the same time, there is no way for us to understand which one is playing a crucial effect.
>
> Ablation study and optimal arm training: Thanks for doing this. I am actually interested in the PreDeToR pre-trained with the expert action (what you did in the paper) vs PreDeToR pre-trained with the optimal action (what you did above I guess?). This helps explain the effect of the optimal action. Can you add one more row?

---

> > ### Author Response · Authors · 2024-12-02
> > **Response to reviewer 5yut**
> >
> > **Additional experiment on Stochastic bandits:** As asked by the reviewer we conducted an additional experiment on K-armed bandits similar to the DPT paper [1]. For the pretraining task distribution, we sample 5 -armed bandits K=5. The reward function for arm $a$ is a normal distribution $\mathcal{N}\left(\mu_a, \sigma^2\right)$ where $\mu_a \sim \operatorname{Unif}[0,1]$ independently and $\sigma=0.3$. To generate in-context datasets $\mathcal{D}_{\text {pre}}$, we randomly generate action frequencies by sampling probabilities from a Dirichlet distribution and mixing them with a point-mass distribution on one random arm. Then we sample the actions accordingly from this distribution. As suggested by the DPT paper, this encourages diversity of the in-context datasets. *It is crucial to note that this has less variance than the all-new action setting we discussed in the paper (Figure 3d)*. We show the result below:
> >
> > | Algorithm  | Regret (n=100) | Regret (n=150) | Regret (n=200) | Regret (n=250) | Regret (n=300) |
> > | ----------------- | ----------- | ------------ | ------------ | ------------ | ------------- |
> > | PreDeToR (Ours)      | 6.2  |      7.9        |     9.19     |     10.58      |    11.91    |
> > | PreDeToR-$\tau$ (Ours)      | 6.06  |     7.2         |   8.97    |     10.13      |  11.0   |
> > |  DPT (Greedy)  | 6.36  |      8.88       |      10.12        |    11.01       |      12.24    |
> > |  DPT (Oracle)  | 5.86  |     7.49         |     8.69         |    9.91       |      10.94    |
> > | Thomp             | 5.56 |     7.33         |     8.6         |     9.19      |   10.93    |
> >
> > Table 2: Cumulative Regret of algorithms at different horizon time in K-armed setting.
> >
> > From the result we see that DPT, PreDeToR-$\tau$ (Ours) matches the performance of Thomp. So PreDeToR-$\tau$ is able to learn a correlation between actions and rewards in the K-armed bandit setting. Again note that PreDeToR-$\tau$ outperforms DPT (Greedy) as it is not able to approximate well the per-task optimal action.
> >
> > **Why DPT perform worse?:** Thanks for understanding the difference between the DPT greedy used and DPT oracle algorithm. We show one additional experiment above to illustrate that PreDeToR-$\tau$ is able to learn a correlation between actions and rewards in the K-armed bandit setting.
> >
> > **Ablation study and optimal arm training:** We are still not sure what is this extra row you are asking for. *We re-iterate in the previous response on 27 Nov, we did not test with PreDeToR-optimal (that is a PreDeToR that knows the optimal action)*. We are not sure how to do this, as we explain again. PreDeToR cannot be trained with the knowledge of optimal action as PreDeToR minimizes the MSE between the observed reward of the *next action and the predicted reward for that action*. Note that the next action selected by the demonstrator may or may not be the optimal action. It is possible you are confused with PreDeToR-$\tau$. In PreDeToR-$\tau$ we introduce a temperature parameter $\tau$ which does an $\epsilon$ greedy type of exploration and generally performs slightly well (see line 244 in draft). We would like to know how the reviewer thinks the PreDeToR-optimal can be trained. We can then quickly run and show the result.

---

### Official Review · Reviewer_hLLK · 2024-11-03

**Soundness:** 3
**Presentation:** 3
**Contribution:** 2
**Rating:** 3
**Confidence:** 4

**Summary:**

This paper introduces a new training phase for a pre-trained Transformer model for structured bandit problems. Unlike existing approaches, such as Algorithm Distillation (AD) and Decision Pretrained Transformer (DPT), which predict the next (near-)optimal action using in-context data during training and testing, the proposed framework focuses on predicting the reward vector and selecting the arm based on this prediction.  This paper gives several experiments and a theoretical analysis of the proposed approach.

**Strengths:**

The paper is well-written. The proposed training framework reduces the dependency on (near-)optimal solutions in training data/tasks, making it more flexible. Additionally, the authors provide a set of experiments and offer a theoretical analysis to understand the framework's effectiveness.

**Weaknesses:**

[1]The primary advantage claimed for the proposed framework is its ability to function without requiring near-optimal solutions in the training data. However, for bandit problems specifically, this requirement is often not a major limitation. In simulated tasks, optimal arms are readily available, or near-optimal actions can be efficiently obtained through established algorithms like UCB or Thompson Sampling. To convincingly illustrate the advantage, experiments on more complex tasks, where identifying near-optimal actions is genuinely challenging, would strengthen the claim.


[2] The proposed framework appears to be limited to situations with a finite set of actions, which restricts its applicability in problems with a continuous action space. Is there a potential adaptation that could allow the framework to handle continuous action spaces? Additionally, the experiments are conducted with a relatively small action space (20 arms), which raises questions about performance in larger action spaces, such as those with over 100 arms.

[3]Although the framework outperforms other benchmarks in most experiments, the assumption that each arm’s features remain constant across training and testing tasks is atypical for structured bandit problems. In standard settings, arm features are generally assumed to be known and vary between tasks. It is unclear why the authors chose this fixed-feature assumption while not using these features as inputs in the transformer model, which seems to reduce the framework’s applicability. Can this assumption be relaxed to better reflect standard structured bandit settings?

[4]When tested with new actions, the framework’s performance declines significantly, as seen in Figure 3(c) and (d). This suggests that the pre-trained Transformer struggles to generalize in cases with out-of-distribution (OOD) actions because the pre-training data lacks these new actions. My question is: if the pre-training can relax the fixed-feature assumption—e.g., by allowing arm features to be randomly sampled in training— would this improve the model’s generalization capabilities?

**Questions:**

Please see the weaknesses. In addition, Line 272 states, “DPT-greedy estimates the optimal arm using the reward estimates for each arm during each task,” which appears to differ from the definition in Section 2.3 and Lee et al. (2023). Could you clarify whether DPT-greedy indeed uses the reward estimate or the best-arm estimate?

---

> ### Author Response · Authors · 2024-11-19
> **Response to Reviewer hLLK**
>
> We thank the reviewer for their reviews. There are several misunderstandings of the reviewer which we wish to clarify.
>
> Weakness:
>
> 1. This is a multi-task structured bandit setting where the structure can be as complex as non-linear/feature bandit to Movielens and Yelp datasets. Typically in a practical recommendation system, the number of rounds of interactions is small. For example, the average rounds of interaction in Movielens is around 10-15. We show that when the number of rounds is less it is difficult for any single task bandit algorithm to perform well and they must leverage the shared structure across tasks. This is where PreDeToR shines as it predicts the next reward of each action and we show that this helps in learning a latent representation that is generalizable to unseen tasks.
>
> 2. We agree that our setting is limited to a finite set of actions. We leave extending this work to a continuous set of actions in future work. We also note that DPT [1], [2] is also for a finite set of actions and a similar number of actions. We experiment with 20-100 number of actions due to computational complexity and limited resources. The attention framework of the transformer requires O(n^2) computational complexity for a context size of n. Our work can be easily extended to a larger set of actions with sufficient resources.
>
> 3. This is a standard assumption in most multi-task representation learning works in structured bandits. We cite a few here and in the paper (see [3]-[7]). We are not aware why the reviewer is raising this point and will be grateful if they can point to some relevant sources. The main goal of our paper is to understand how the PreDeToR leverages shared latent representation across tasks. The shared structure lets the transformer learn a correlation between tasks, the action selected, and the underlying feedback. The [1] and [2] papers are a few examples that exactly study the same setting as ours.
>
> 4. We do not fully follow the reviewer. We stated in lines 408-411 that we introduce new arms both during train and test time. However, this results in the breaking down of the shared structure assumption and we show that PreDeToR is not able to preform well. So PreDeToR does not generalize well to OOD. This is exciting future work we aim to study. We hope this clears the doubt of the reviewer on the importance of the shared structure.
>
> Questions:
> 1. We apologize for the confusion. The original DPT paper has the oracle access to the optimal action. This is what we state in section 2.3. The DPT greedy estimates the optimal arm using the reward estimates for each arm during each task. This DPT greedy is what we use in our experiments as this is a more practical algorithm.
>
>
> [1] Supervised Pretraining Can Learn In-Context Reinforcement Learning, Jonathan N. Lee, Annie Xie, Aldo Pacchiano, Yash Chandak, Chelsea Finn, Ofir Nachum, Emma Brunskill.
>
> [2] Transformers as Statisticians: Provable In-Context Learning with In-Context Algorithm Selection, Yu Bai, Fan Chen, Huan Wang, Caiming Xiong, Song Mei
>
> [3] Multi-task Representation Learning with Stochastic Linear Bandits, Leonardo Cella, Karim Lounici, Grégoire Pacreau, Massimiliano Pontil
>
> [4] Multi-task Representation Learning for Pure Exploration in Linear Bandits, Yihan Du, Longbo Huang, Wen Sun
>
> [5] Nearly Minimax Algorithms for Linear Bandits with Shared Representation, Jiaqi Yang, Qi Lei, Jason D. Lee, Simon S. Du
>
> [6] Impact of Representation Learning in Linear Bandits, Jiaqi Yang, Wei Hu, Jason D. Lee, Simon S. Du
>
> [7] Multi-task representation learning for pure exploration in bilinear bandits, Subhojyoti Mukherjee, Qiaomin Xie, Josiah Hanna, Robert Nowak

---

> > ### Author Response · Authors · 2024-11-26
> > **Response and summarize our view point for reviewer hLLK**
> >
> > Hi Reviewer hLLK,
> >
> > Since the discussion time is coming to an end we wanted to summarize our viewpoint and still hope that you will engage in discussion.
> >
> > 1. **Multi-task structured bandit setting**: We study multi-task structured bandit setting where we show that learning the underlying latent representation is crucial in performing optimally (reducing cumulative regret). Near-optimal actions cannot be approximated well enough in this setting without learning the shared structure and so DPT fails.
> > 2. **Shared structure is common in MTRL**: The shared structure is a common assumption in all multi-task representation learning settings. We shared several related works [1]-[7] in the paper and above response. The main focus of this paper is to show that PreDeToR learns the underlying latent shared structure and the feedback function as well.

---

> > > ### Comment · Area_Chair_E1CA · 2024-11-30
> > > **Please respond to the authors**
> > >
> > > Dear Reviewer hLLK,
> > >
> > > Please respond to the authors.
> > >
> > > Thanks!
> > >
> > > The AC

---

### Official Review · Reviewer_6SDC · 2024-11-04

**Soundness:** 2
**Presentation:** 3
**Contribution:** 2
**Rating:** 6
**Confidence:** 4

**Summary:**

This paper introduces a new supervised pretraining procedure to learn the underlying reward structure in a multi-task structured bandit setting, which can be particularly useful when the optimal action is unknown in the training phase. This training method leads to improved performance across various bandit tasks, including linear, nonlinear, bilinear, and latent bandits. The authors also provide a theoretical generalization bound for the proposed algorithm.

**Strengths:**

I think the idea of predicting reward function rather than the optimal action has its merit. The numerical experiment is thorough, and the intuitions are well-explained.

**Weaknesses:**

I mainly have two concerns of the paper:

A distribution shift issue that doesn't arise under the in-context learning setting:
- The paper draws the analogy between the pretrained decision transformer with in-context learning multiple times throughout the paper, for both algorithm design and the theoretical bound. For in-context learning problems, we can think the task from the pretraining phase as being i.i.d. as the task during the test phase. But for the bandits problem, things are quite different. Specifically, the actions in $H_m$ generated during the pre-training phase is from weak demonstrators, whereas the actions in the test phase are generated from the transformer model itself. This will result in a distribution shift between the training and the test environment. And it matters both numerically and theoretically:
- Numerically, if the transformer's policy takes actions similar to the demonstrators such as UCB and TS, then why would it has a better regret performance than them? If the transformer's policy doesn't take actions similar to the demonstrators such as UCB and TS, how does the training through samples of $H_m$ from demonstrators extrapolate to the cases where the actions in $H_m$ is sampled from the transformer's policy? Given the large space of $H_m$, I don't believe the samples of $H_m$ following demonstrators can cover the whole space.
- Theoretically, I don't see any discussion of this distribution shift matter in the theoretical results in Section 7 (please correct me if I miss anything). To derive generalization bound in the face of this distribution shift, I believe notions such as the distribution ratio in the definition 5 of  https://arxiv.org/pdf/2310.08566 or others are necessary.

Medium- to large-size action space or continuous action space: As I read from the algorithm and the experiments, the proposed method does not well handle tasks with continuous actions or with a large action space well, which are typical in linear and bilinear bandits. For tasks with a large number of actions, the proposed method becomes challenging to train because it requires H_m to include some reward for every possible action.

**Questions:**

See above.

---

> ### Author Response · Authors · 2024-11-19
> **Response to Reviewer 6SDC**
>
> We thank the reviewer for their feedback. We answer below:
> 1. We agree that the distribution of the input contexts will shift. However, as shown in the experiments, we have not found this to be an issue in practice. Moreover, even though there is a distribution shift, the hidden structure means that the pre-training data covers the space better than if there was no hidden structure. Finally, we show that when the hidden structure breaks down because of the distribution shift the PreDeToR (and DPT/AD) fails to perform well. Note that the [1] and [2] papers are a few examples that exactly study the same setting as ours.
> 2. Yes, UCB and TS are weak demonstrators that do not leverage the structure. However, the transformer with sufficient model complexity can leverage the shared structure across the tasks. The main goal of our paper is to understand how the PreDeToR leverages shared latent structure across tasks given the in-context data with no feature information. The shared structure lets the transformer learn a correlation between tasks, the action selected, and the underlying feedback. The [1] and [2] papers are few examples that exactly study the same setting as ours.
>
>    [1] fails in settings where the number of interactions is less because it needs the optimal action or a greedy approximation of it. In contrast, PreDeToR predicts the next reward of each action and we show that this helps in learning a latent representation that is generalizable to unseen tasks. This is best illustrated with this simple example. Consider two users: User 1, who prefers item 1 over item 2, and User 2 who prefers item 2 over item 1. We have an interaction history of many such users. Then we have a new user 3 with no past interaction history. After observing a few in-context examples/feedback the transformer is able to figure out which item to suggest to the user. We discuss this exploration in detail in Appendix A.11 and also extend the exploration study to unseen items in Appendix A.12.
>
> 3. Yes, theoretically we do not show a connection to distribution shift.  Note that PreDeToR minimizes an MSE loss, and the PreDeToR algorithm is a greedy algorithm that selects the action with the highest predicted reward for the test task (after observing a few samples from the test task). So we show how PreDeToR generalizes in identifying the test task after a few in-context samples from the test task. This line of research follows the work of [4] who also draw a connection as to how the transformer algorithm can generalize in context. We leave the work of showing the influence of distribution shift in our bound to future work using the theoretical understanding of  [3].
>
> [1] Supervised Pretraining Can Learn In-Context Reinforcement Learning, Jonathan N. Lee, Annie Xie, Aldo Pacchiano, Yash Chandak, Chelsea Finn, Ofir Nachum, Emma Brunskill.
>
> [2] Transformers as Statisticians: Provable In-Context Learning with In-Context Algorithm Selection, Yu Bai, Fan Chen, Huan Wang, Caiming Xiong, Song Mei
>
> [3] Transformers as Algorithms: Generalization and Stability in In-context Learning, Yingcong Li, M Emrullah Ildiz, Dimitris Papailiopoulos, Samet Oymak
>
> [4] Transformers as decision-makers: Provable in-context reinforcement learning via supervised pertaining, Licong Lin, Yu Bai, Song Mei

---

> ### Author Response · Authors · 2024-11-26
> **Response and summarize our view point for reviewer 6SDC**
>
> Hi Reviewer 6SDC,
>
> Since the discussion time is coming to an end we wanted to summarize our viewpoint and still hope that you will engage in discussion.
>
> 1. **Distribution shift:** This is not an issue when the structure is shared between the tasks. When the structure breaks down we show in section 6, "EMPIRICAL STUDY: IMPORTANCE OF SHARED STRUCTURE" that DPT, AD and PreDeToR fails.
> 2. **Better performance than demonstrator**: The UCB and TS demonstrators do not leverage the shared structure in the multi-task setting. We show that PreDeToR and DPT do leverage the shared structure. The multi-task linear regression algorithm MLinGreedy leverages the shared structure but fails in non-linear settings.

---

> > ### Comment · Area_Chair_E1CA · 2024-11-30
> > **Please respond to authors**
> >
> > Dear Reviewer 6SDC,
> >
> > Please respond to the authors.
> >
> > Thanks!
> >
> > The AC

---

### Author Response · Authors · 2024-11-22
**Message to Area chair**

Hi Area chair,

We have put significant effort into the paper and there are some serious misunderstandings on the part of the reviewers which we again point out below. It will be great if you urge some discussion on their part.

1. Reviewer 5yut: We have pointed out that we indeed discuss in detail how PreDeToR conducts a two-phase exploration in Appendix A.11 and Appendix A.12 which has not been shown by previous works. In fact, reviewer 5yut has not read the work properly and has asked us to cite the DPT paper which actually has been cited many times in the draft (and is our closest baseline).

2. Reviewer hLLK: The reviewer has a misunderstanding of how multi-task representation learning works for structured/linear bandit settings. We pointed out several works that have the same setting as ours. The reviewer has a misunderstanding on what we are showing in OOD generalization. We argue that all the current baselines (including ours) fail in the OOD setting, and shared structure is crucial in learning the optimal algorithm. This is an insight from our work. This has been counted as a weakness which is not a correct evaluation of the work.

We have made this comment public to everyone so it can lead to some discussion.

---

### Meta-Review · Area_Chair_E1CA · 2024-12-22

**Metareview:**

In this work the authors study how to solve multi-task structured bandit problems via a meta learning procedure with similarity to the decision pretrained transformer. The main modification of this work is that in contrast with DPT, in this work the authors propose a method based on predicting the rewards of query arms. This means it this method does not require optimal arm supervision. It is also shown the method can learn the structure of the reward class such as when these are linear.

I like the direction of this work, but it is a bit hard to argue for acceptance given the reviewer’s comments. I went over the author’s comments and the discussion with the reviewers and I also looked at the paper myself. One of the main issues of this work is presentation. It would help the authors to present their work in and slightly easier to read, by describing their algorithms as a meta-learning procedure, where the objective is to learn a good bandit history dependent reward predictor by pretraining on a distribution of source bandit tasks. These predictors can be used to construct something that looks like a Thompson sampling algorithm. I am somewhat worried that the proposed strategies, PreDeToR and PreDeToR-tau do not have probable exploration schemes. Both greedy and softmax strategies may not yield provable exploration guarantees. Another issue is the experimental evaluation. The reviewers were worried that the scenarios discussed in this work are limited. I understand the authors replicated many of the DPT experimental scenarios and I sympathize with that. Nonetheless having multiple readers of the work point out to the limited experimental results, does represent a signal that more experiments could be conducted.

**Additional Comments On Reviewer Discussion:**

As detailed above, two issues were raised by reviewers. First, it was hard for reviewers to understand the setup of the paper. This could point to the work requiring an improvement in the writing. Second, the reviewers agreed the experimental evaluation does not contain enough information to understand the significance of the paper's results. The rejection recommendation was tough to get to.

---

### Decision · Program_Chairs · 2025-01-22

Reject